# Transcription factor EB regulates phosphatidylinositol-3-phosphate levels that control lysosome positioning in the bladder cancer model

Pallavi Mathur[1,2,3], Camilla De Barros Santos[1,2], Hugo Lachuer [1,2,3], Julie Patat[3,4], Bruno Latgé[1,2], François Radvanyi [1,2], Bruno Goud [1,2] & Kristine Schauer [1,2,3,4 ✉]

Lysosomes orchestrate degradation and recycling of exogenous and endogenous material thus controlling cellular homeostasis. Little is known how this organelle changes during cancer. Here we investigate the intracellular landscape of lysosomes in a cellular model of bladder cancer. Employing standardized cell culture on micropatterns we identify a phenotype of peripheral lysosome positioning prevailing in bladder cancer cell lines but not normal urothelium. We show that lysosome positioning is controlled by phosphatidylinositol-3-phosphate (PtdIns3P) levels on endomembranes which recruit FYVE-domain containing proteins for lysosomal dispersion. We identify transcription factor EB (TFEB) as an upstream regulator of PtdIns3P production by VPS34 that is activated in aggressive bladder cancer cells with peripheral lysosomes. This conceptually clarifies the dual role of TFEB as regulator of endosomal maturation and autophagy, two distinct processes controlled by PtdIns3P. Altogether, our findings uncover peripheral lysosome positioning, resulting from PtdIns3P production downstream of TFEB activation, as a potential biomarker for bladder cancer.

[1] Centre National de la Recherche Scientifique, UMR144, 75005 Paris, France. [2] Institut Curie, PSL Research University, 75248 Paris, France. [3] Institut Gustave Roussy, INSERM UMR1279, 94805 Villejuif, France. [4] Paris-Saclay University, 91190 Gif-sur-Yvette, France. ✉email: kristine.schauer@gustaveroussy.fr

Accelerated cellular division and enhanced motility are pathological characteristics of malignant cells both leading to an increase in energetic demand. More than being the "stomach" of eukaryotic cells for nutrient acquisition, late endosomes/lysosomes (referred to as lysosomes hereafter) have emerged as a cellular hub for metabolism and signaling[1–5] and play an important role during cancer development[3,6]. Lysosomes are morphologically heterogeneous acidic compartments that are functionally similar to yeast and plant vacuoles. They are specialized in the degradation of extracellular molecules and pathogens internalized by endocytosis or phagocytosis, as well as the intracellular recycling of macromolecules and organelles sequestered by autophagy. In addition to the orchestration of cellular clearance, lysosomes play an important role in cellular nutrient availability controlled by the mammalian target of rapamycin complex 1 (mTORC1)[7]. Active mTORC1 assembles at the surface of lysosomes through the integration of chemically diverse nutrient and growth factor signaling to promote protein biosynthesis[1,5,7]. Conversely, absence of nutrients triggers the dissociation and inactivation of mTORC1 and consequently the activation of downstream catabolic pathways. Active mTORC1 targets MiT/TFE transcription factors, including transcription factor EB (TFEB) and MITF, that are both master regulators of lysosome biogenesis and autophagy[8]. MiT/TFE transcription factors have been implicated in the development of cancer, including renal cell carcinoma, pancreatic adenocarcinoma, sarcoma and melanoma, MITF being an important oncogene in melanoma[4]. Moreover, it has been shown that TFEB overexpression as well as a positive feedback mechanism between mTORC1 and TFEB was sufficient to promote cancer growth in mouse models[9,10].

Although lysosomes are important for nutrient acquisition and the regulation of metabolism, both prerequisites for malignant growth, little is known how lysosomes change during cancer development. Here, we compare the intracellular landscape of the lysosomal compartment in a collection of bladder cancer cell lines to normal human urothelium (NHU). Bladder cancer represents one of the most frequently-diagnosed cancer types worldwide and is among the most common neoplasms in men in North America and Europe, thus representing an important health burden[11]. Bladder carcinomas are highly diverse and are classified into non-muscle-invasive bladder cancers (NMIBC) and muscle-invasive bladder cancers (MIBC) with luminal-like and basal-like subtypes[12,13]. Investigating the normal and pathologic landscape of lysosome positioning in cells representing different stages of bladder cancer, we here reveal organelle-level deregulation in malignant cells and identify TFEB as major regulator of phosphatidylinositol-3-phosphate (PtdIns3P) homeostasis in this context.

## Results

### Cell lines representing high-grade bladder cancers are characterized by a peripheral positioning of lysosomes.

Because of the importance of lysosomes in cellular homeostasis and their role in promoting cancer progression, we aimed at a systematic analysis of lysosome morphology in a panel of genetically diverse bladder cancer cell lines in comparison to primary normal human urothelium (NHU) cells. We have analyzed the broadly studied bladder cancer cell lines RT4, MGHU3, RT112, KU19-19, T24, TCCSup and JMSU1 that represent the diversity of bladder carcinomas[14]. RT4, MGHU3, RT112 represent low-grade, luminal cancers of the papillary subtype, whereas KU19-19 represents high-grade, basal cancers and T24, TCCSup and JMSU1 represent high-grade cancers of mixed subtypes[14,15]. To compare these different cells at the morphological level, we cultured them on identical crossbow-shaped micropattern substrates. All tested cells were fully spread after 3 h of incubation, visualized by the average projection of the actin cytoskeleton (Fig. S1A), indicating that all cells adapted well to the micropatterns. We visualized the lysosomal compartment in all cells by immunofluorescence staining of the lysosomal-associated membrane protein 1 (LAMP1/CD107a) (Fig. 1A). Images were acquired in 3D and lysosomes were segmented to obtain quantitative information of their spatial organization, volume and numbers per cell. To visualize the average lysosome organization, we plotted 3D density maps[16,17] representing the smallest cellular volume that contains 50% of lysosomes (Fig. 1B). Notably, while in NHU cells lysosomes were positioned centrally, they were found to be spread out to the periphery in cancerous cells with the strongest phenotype exhibited in cell lines representing high-grade cancers (Fig. 1A, B). Because the total cell area is standardized by the micropattern and thus identical in all cells, we calculated the nearest neighbor distance (NND) of lysosomes in each cell. Concomitantly, whereas the average NND in RT4 and MGHU3 cells was not significantly different from NHU cells, those of all other analyzed cell lines was significantly increased (Fig. 1C), indicating that lysosomes are more scattered in these cells. No clear trend in the number of lysosomes per cell (Fig. 1D) or average volume (Fig. 1E) was found among the tested cell lines. However, lysosomal volume negatively correlated with lysosomal number (Fig. S1B), indicating that few large lysosomes are in balance with many small ones. Principal component analysis (PCA) of the transcriptome data of these cells indicated that replicates of the NHU clustered together and separately, and that RT4 and MGHU3 were the most different compared to the other cell lines (Fig. S1C). Comparison between MGHU3 (luminal-type and central lysosomes), RT112 (luminal-type and scattered lysosomes), KU19-19 (basal-type) and JMSU1 cells (mixed-type) in invasion assays into collagen matrix from spheroids revealed, as expected, that MGHU3 and RT112, representing luminal cancers, are less invasive than KU19-19 or JMSU1 that represent high-grade cancers. MGHU3 was the less invasive cell line (invasion at 5 d), followed by RT112 (invasion at 3 d), whereas KU19-19 and JMSU1 both invaded at 1 d with different efficiency (Fig. S1D). To verify that changes in lysosomal positioning were not induced by micropatterning, we additionally analyzed lysosomes in classical cell culture conditions in these cell lines. We measured the averaged squared distance of lysosomes to the center of mass of the cell (statistical inertia) normalized to the cell size (Fig. S1E, F). In agreement with our density map and NND analysis, the lysosome dispersion significantly increased from MGHU3 to JMSU1 cells. Our analyses collectively indicate that the lysosomal compartment shows differences between NHU and bladder cancer cell lines. Whereas some cell lines representing low-grade bladder cancers reveal central lysosomes similar to NHU cells, cell lines representing high-grade bladder cancers are characterized by a scattered, peripheral positioning of the lysosomal compartment.

### Dispersed lysosomes reveal alterations in the mTORC1-TFEB signaling axis.

Lysosomes are the cellular signaling platform for the mammalian target of rapamycin complex 1 (mTORC1), a main regulator of metabolisms, proliferation and survival. Because mTORC1 is regulated by lysosomes positioning[3,18], we tested whether altered lysosome landscape across different bladder cancer cell lines affected mTORC1 signaling. First, we analyzed mTORC1 localization by co-visualizing mTOR and LAMP1 by immunofluorescence and measuring the fraction of mTOR that localized on the LAMP1-positive compartment. We found that about 15–20% of mTOR signal was associated

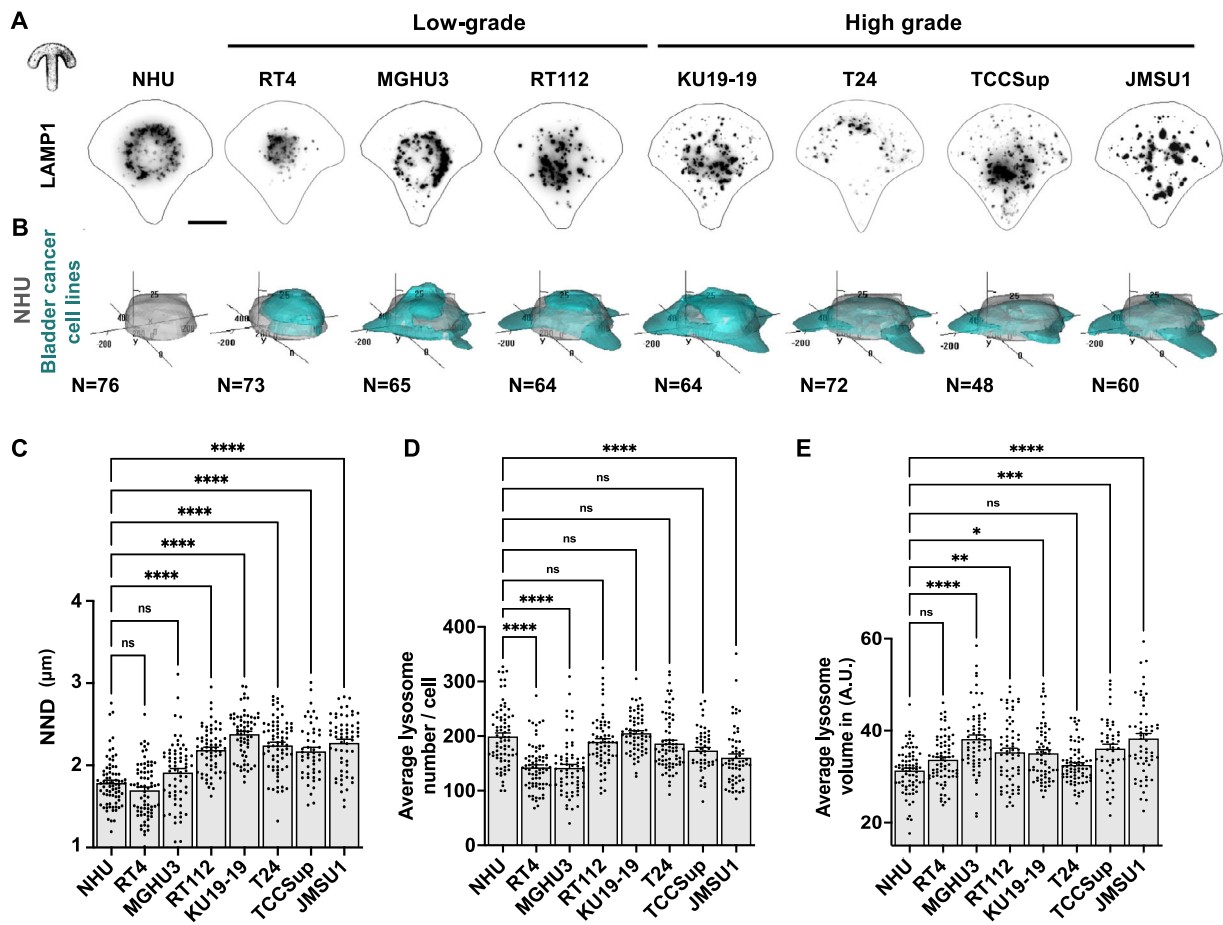

**Fig. 1 High-grade cancer cell lines are specifically characterized by scattered, peripheral positioning of lysosomes. A** Representative images of lysosomes visualized by immunofluorescence staining against the lysosomal-associated membrane protein 1 (LAMP1/CD107a) in normal human urothelium (NHU) and bladder cancer cell lines RT4 (ATCC® HTB-2™), MGHU3[70], RT112[71], KU19-19[72], T24, TCCSup[73], JMSU1[74] cells cultured on crossbow-shaped adhesive micropatterns for better comparison. Scale bar is 10 μm. **B** 3D probabilistic density maps of lysosomes of N cells of NHU, RT4, MGHU3, RT112, KU19-19, T24, TCCSup and JMSU1. The 50% contour visualizes the smallest cellular volume containing 50% of lysosomes. **C** Nearest neighbor distance (NND) between lysosomes in NHU ($n = 76$), RT4 ($n = 73$), MGHU3 ($n = 65$), RT112 ($n = 64$), KU19-19 ($n = 64$), T24 ($n = 72$), TCCSup ($n = 48$) and JMSU1 ($n = 60$). Adjusted $p$ values of testing against NHU condition are RT4: > 0.9999, MGHU3: 0.1943; RT112: < 0.0001; KU19-19: <0.0001; T24: < 0.0001; TCCsup: <0.0001; JMSU1: < 0.0001 in a Kruskal-Wallis test with Dunn's test for multiple comparisons; ns $p > 0.1$ and ****$p < 0.0001$, error bars are SEM. **D** Average numbers of lysosomes per cell in NHU ($n = 76$), RT4 ($n = 73$), MGHU3 ($n = 65$), RT112 ($n = 64$), KU19-19 ($n = 64$), T24 ($n = 72$), TCCSup ($n = 48$) and JMSU1 ($n = 60$). Adjusted $p$ values of testing against NHU condition are RT4: <0.0001; MGHU3: < 0.0001; RT112: > 0.9999; KU19-19: 0.8807; T24: > 0.9999; TCCsup: 0.2068; JMSU1: < 0.0001 in a Kruskal-Wallis test with Dunn's test for correction for multiple comparisons; ns $p > 0.1$ and ****$p < 0.0001$, error bars are SEM. **E** Average volume of lysosomes in NHU ($n = 76$), RT4 ($n = 73$), MGHU3 ($n = 65$), RT112 ($n = 64$), KU19-19 ($n = 64$), T24 ($n = 72$), TCCSup ($n = 48$) and JMSU1 ($n = 60$). Adjusted $p$ values of testing against NHU condition are RT4: 0.1414; MGHU3: < 0.0001; RT112: 0.0048; KU19-19: 0.0110; T24: > 0.9999; TCCsup: 0.0003; JMSU1: < 0.0001 in a Kruskal-Wallis test with Dunn's test for multiple comparisons; ns $p > 0.1$, *$p < 0.1$, ***$p < 0.001$ and ****$p < 0.0001$, error bars are SEM.

with lysosomes. Although RT112 showed slightly but significantly more mTOR on lysosomes, the levels of mTOR on lysosomes were comparable between the tested cell lines (Fig. 2A, B). Next, we tested mTORC1 activity monitoring the phosphorylation of the direct downstream substrates p70-S6 Kinase 1 (S6K1) and eIF4E Binding Protein (4EBP1) that are phosphorylated during activation of protein synthesis. We found both substrates were phosphorylated in bladder cancer cells (Fig. 2C, D), with particularly strong phosphorylation of S6K1 in MGHU3 and RT112 cells and strong phosphorylation of 4EBP1 in JMSU1 cells. Whereas total S6K1 levels were similar in all cell lines, 4EBP1 expression was also increased in JMSU1 cells (Fig. S2A, B). As expected, the mTORC1 inhibitor rapamycin[19,20] as well as starvation decreased S6K1 and 4EBP1 phosphorylation in all cell lines confirming mTORC1 specificity (Fig. S2C–E).

Next, we tested another important mTORC1 substrate, the transcription factor EB (TFEB), which appears as a novel player in carcinogenesis[9,10]. Because phosphorylation of TFEB retains this transcription factor in the cytosol, whereas the active form is nuclear, we transfected cells with TFEB-EGFP and monitored its localization in cells 72 h post transfection. Whereas TFEB-EGFP showed cytosolic localization in MGHU3 and RT112 cells (40% of mean TFEB-EGFP intensity was found in nucleus), more than 70% of the mean intensity of TFEB-EGFP was found in the nucleus of KU19-19 and JMSU1 cells (Fig. 3A, B). To confirm the results of overexpressed TFEB-EGFP, we performed cell fractionation and separated the nuclear and cytosolic fractions, in which endogenous TFEB was monitored by western blotting. We used LaminB, a component of the nuclear envelope, as the nuclear marker and GAPDH as the cytosolic marker to ensure the efficiency of fractionation. Consistent with the TFEB-EGFP

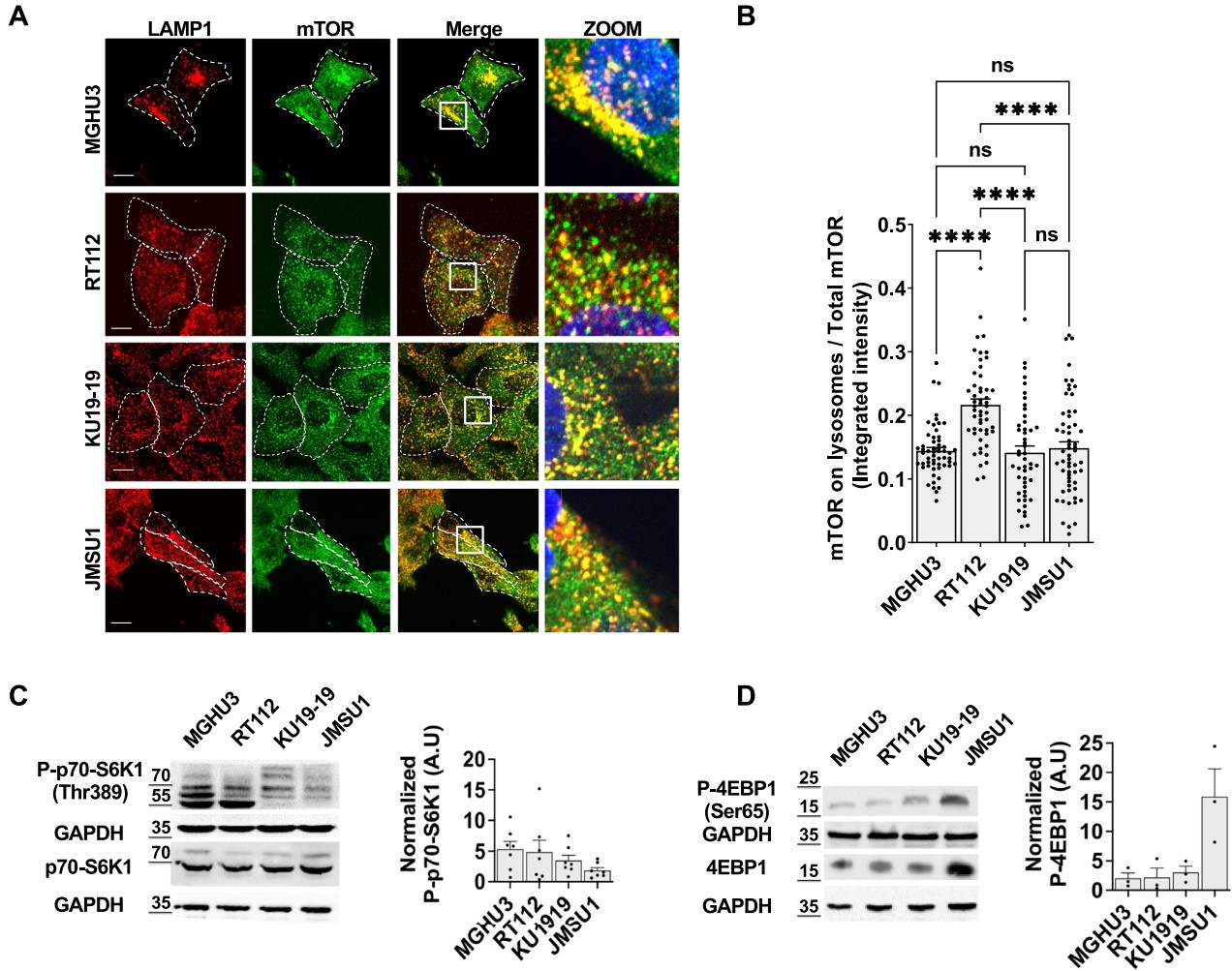

**Fig. 2 Dispersed lysosomes reveal alterations in mTORC1 signaling. A** Immunofluorescence staining of the lysosomal-associated membrane protein 1 (LAMP1, CD107a) and mTOR in MGHU3, RT112, KU19-19 and JMSU1. The zoom shows the merged image for both proteins in the white box. Scale bars equal 15 μm **B** Quantification of mTOR intensity on lysosomes normalized to total cellular mTOR approximately 50 cells for each cell line; ****$p < 0.0001$ in a Kruskal-Wallis test with Dunn's test for multiple comparison, error bars are SEM. **C** Western blot analysis of phosphorylated p70-S6 Kinase 1 (P-p70-S6K1 Thr389) and total p70-S6K1 as well as GAPDH loading control in MGHU3, RT112, KU19-19 and JMSU1 cells and quantification of P-p70-S6K1 levels normalized to GAPDH from $n = 7$ experiments, error bars are SEM, molecular weight markers are in kDa. **D** Western blot analysis of phosphorylated 4EBP1 (P-4EBP1 Ser65) and total 4EBP1 as well as GAPDH loading control in MGHU3, RT112, KU19-19 and JMSU1 cells and quantification of P-4EBP1 levels normalized to GAPDH from $n = 3$ experiments, error bars are SEM, molecular weight markers are in kDa.

results, we observed that TFEB was predominantly found in the nucleus in JMSU1 cells but not in RT112 cells (Fig. 3C, D). Notable, nuclear TFEB appeared to have a higher molecular weight than cytosolic TFEB in bladder cancer cell lines. This observation opens the question whether post-translational modifications, such as acetylation, SUMOylation, PARsylation, or glycosylation in addition to phosphorylation[21], or alternatively, different isoforms of TFEB could be more prevalent in the nuclear fraction of bladder cancer cells. To further test whether this nuclear localization indicated hyperactivation of TFEB in cell lines representing high-grade cancers we performed a GSEA (Gene Set Enrichment Analysis) of the transcriptome of bladder cancer cell lines. GSEA analysis demonstrated that genes belonging to the TFEB-regulated Coordinated Lysosomal Expression and Regulation (CLEAR) network[22,23] are upregulated in cell lines representing high-grade bladder cancers as compared to cell lines representing low-grade bladder cancers (Fig. 3E) supporting the hypothesis that TFEB is active. Indeed, inspection of known TFEB-regulated genes such as *RRAGD*, encoding RagD and overexpressed in tumors[10], or *TSC1* revealed an increase in their

expression in KU19-19 and JMSU1 cells (Fig. S3A) highlighting potential deregulation in TFEB signaling.

Next, we investigated mechanisms underlying differential TFEB localization in RT112 and JMSU1 cells. Interestingly, addition of rapamycin or starvation in RT112 cells led to a translocation of TFEB-EGFP into the nucleus (Fig. 3F, G and S3B, C). Rapamycin-sensitivity of TFEB and nuclear translocation was previously observed in cells with highly expressed lysosome calcium channel, mucolipin-1, also known as transient receptor potential channel mucolipin 1 (TRPML1 or MCOLN1)[24]. Rapamycin can directly activate mucolipin-1[24] that results in TFEB dephosphorylation by calcium-dependent activation of the phosphatase calcineurin[25]. However, inhibition of mucolipin-1 using GW-405833 (ML-SI1, 2 h) in the presence of rapamycin in RT112 cells did not prevent nuclear translocation of TFEB-EGFP (Fig. 3F, G). Contrary, inhibiting mucolipin-1 in JMSU1 cells with predominantly nuclear TFEB using GW-405833 (ML-SI1) or incubating cells with the calcium chelator BAPTA for 2 h each led to the cytoplasmic translocation of TFEB-EGFP (Fig. 3H, I). This indicated that lysosomal calcium release through mucolipin-1

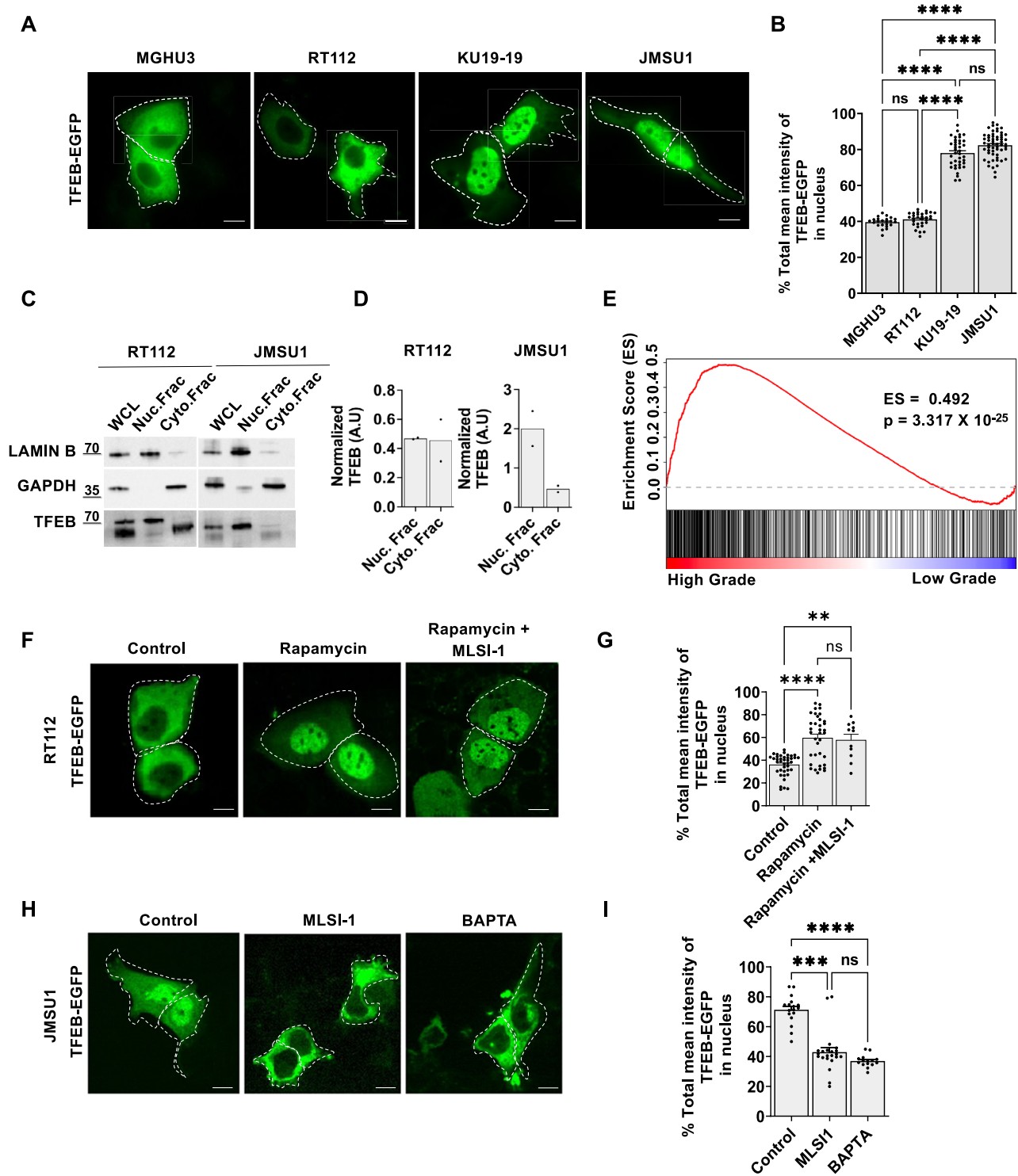

contributes to the nuclear accumulation of TFEB-EGFP in JMSU1 cells, but not in RT112 cells. Together, our results indicate that mTORC1 is active in all bladder cancer cell lines analyzed, despite a nuclear localization of TFEB in KU19-19 and JMSU1 cells that correlates with an activation of TFEB-regulated CLEAR network genes.

**Lysosome positioning correlates with TFEB activation in bladder cancer cells.** It has been previously reported that TFEB regulates lysosomal positioning[26], thus, we investigated whether increased nuclear translocation of TFEB in bladder cancer cell lines could lead to peripheral lysosomes. First, we tested whether stimulating nuclear translocation of TFEB in RT112 cells triggered anterograde lysosome movement. Thus, RT112 cells were either treated with rapamycin or were starved for 24 h and lysosomes were visualized by immunofluorescence against LAMP1. Inspection of classically cultured cells revealed recurrent accumulation of lysosomes at the cell periphery upon rapamycin treatment (Fig. 4A) or starvation (Fig. S4A). To quantify this, we

**Fig. 3 TFEB regulation in bladder cancer cell lines. A** Representative images of MGHU3, RT112, KU19-19 and JMSU1 cells transfected with TFEB-EGFP for 72 h. Scale bars equal 10 µm. **B** Quantification of the nuclear fraction of the total mean TFEB-EGFP fluorescent intensity in MGHU3 ($n = 23$), RT112 ($n = 31$), KU19-19 ($n = 39$) and JMSU1 cells ($n = 57$); ns $p > 0.05$, ****$p < 0.0001$ in a Kruskal-Wallis test with Dunn's test for multiple comparison. Error bars are SEM. **C** Western blot analysis of endogenous TFEB in RT112 and JMSU1 cells after cell fragmentation in fractions of whole cell lysate (WCL), nuclear fraction (Nuc.Frac) and cytosolic fraction (Cyto.Frac) with LaminB as the marker of the nuclear fraction and GAPDH as the marker of cytosolic fraction, molecular weight markers are in kDa. **D** Western blot quantification of endogenous TFEB in RT112 and JMSU1 cells after cell fragmentation in nuclear fraction (Nuc.Frac) and cytosolic fraction (Cyto.Frac) normalized to TFEB in whole cell lysate (WCL) from $n = 2$ experiments. **E** Gene Set Enrichment Analysis (GSEA) results showing enrichment score (ES) of TFEB regulated CLEAR network genes in bladder cancer cell lines representing high-grade (JMSU1, KU19-19, T24, TCCSup) vs low-grade (RT4, MGHU3, RT112). **F** Representative images of RT112 cells transfected with TFEB-EGFP for 72 h in control (DMSO) and treated with 10 µM rapamycin and rapamycin + ML-SI1 for 2 h. Scale bars equal 10 µm. **G** Quantification of the nuclear fraction of the total mean TFEB-EGFP fluorescent intensity in control (DMSO), and rapamycin (2 h) and rapamycin + ML-SI1 (2 h) conditions (for $n > 10$ cells in each condition). ns $p > 0.05$, ****$p < 0.0001$, **$p < 0.01$ in a Kruskal-Wallis test with Dunn's test for multiple comparison, error bars are SEM. **H** Representative images of JMSU1 cells transfected with TFEB-EGFP for 72 h in control (DMSO) and treated with ML-SI1 or BAPTA AM for 3 h. Scale bars equal 10 µm. **I** Quantification of the nuclear fraction of the total mean TFEB-EGFP fluorescent intensity in control (DMSO), ML-SI1 (2 h) and BAPTA AM (2 h) treatment conditions (for >15 cells in each condition); ns $p > 0.05$, ***$p < 0.001$ and ****$p < 0.0001$ in Kruskal-Wallis test with Dunn's test for multiple comparison, error bars are SEM.

cultured cells on adhesive micropatterns and calculated the nearest neighbor distance (NND) of lysosomes in RT112 cells. Lysosomes were more dispersed after rapamycin treatment in micropatterned cells (Fig. 4B) and the average NND was significantly increased as compared to untreated controls (Fig. 4C). Interestingly, we also observed an increase in the number of lysosomes per cell after rapamycin treatment, potentially indicating activation of lysosomal biogenesis (Fig. S4B). We next depleted TFEB via RNA interference in JMSU1 cells, where TFEB was mostly nuclear. Silencing of TFEB by either a pool of four siRNAs or four individual siRNAs significantly reduced TFEB protein levels after 3 d and reversed the scattered lysosome phenotype in JMSU1 cells (Fig. 4D and S4C–E). Quantification on micropatterns revealed a significant decrease in the average NND of lysosomes (Fig. 4E, F).

It has been shown that lysosomes translocate to the cell periphery upon overexpression of protrudin, and conversely, cluster perinuclearly upon protrudin depletion[27]. Thus, we next tested whether recruitment of protrudin to lysosomes is TFEB-dependent. Because protrudin is an ER-localized protein and only is found on lysosomes at ER-lysosome contact sites, we measured the fraction of protrudin that is found on LAMP1-positive lysosomes. We found increased recruitment of protrudin upon both rapamycin treatment or starvation in RT112 cells (Fig. 4G, H and S4F, G). However, this recruitment was TFEB-independent, because targeting TFEB by siRNA prior to rapamycin treatment or in control conditions did not decrease protrudin levels on lysosomes (Fig. 4G, H and S4H). Total protein levels of protrudin did also not change in RT112 cells after rapamycin treatment, starvation or when TFEB was targeted by siRNA (Fig. S4H, I). Contrary, depletion of TFEB by siRNA in JMSU1 cells significantly reduced protrudin levels on LAMP1-positive lysosomes (Fig. 4I, J), although again the total protein levels of protrudin remained unchanged (Fig. S4J). Note that protrudin gene expression was not up-regulated in bladder cancer cell lines (Fig. S4K). Overall, this suggested that rapamycin and starvation induced lysosomal anterograde movement through protrudin recruitment that was TFEB-independent in RT112 cells. Contrary, in JMSU1 cells, protrudin recruitment and lysosome positioning were under the control of TFEB, but not through the upregulation of protrudin protein expression.

**TFEB regulates phosphatidylinositol-3-phosphate levels on endomembranes in bladder cancer cells.** Recruitment of protrudin to lysosomes is regulated by the binding of its FYVE domain to phosphatidylinositol-3-phosphate (PtdIns3P) found on endomembranes[27]. We thus investigated whether TFEB could regulate lysosomal PtdIns3P levels. We expressed the PtdIns3P-binding FYVE domain from the human homolog of the hepatocyte

growth factor-regulated tyrosine kinase substrate Hrs, duplicated in tandem as an EGFP fusion construct (EGFP-FYVE)[28] and monitored total level of this construct on LAMP1-positive lysosomes upon knock down of TFEB in JMSU1 cells. PtdIns3P is found on early and late endosomes, thus as expected, EGFP-FYVE showed an endosomal staining but only partially colocalized with lysosomes (Fig. 5A). Treatment of EGFP-FYVE expressing cells with wortmannin to inhibit PtdIns3P production by phosphatidylinositol-3-phosphate kinases significantly reduced the levels of EGFP-FYVE on LAMP1-positive lysosomes, confirming the specific binding of this construct to PtdIns3P on lysosomes (Fig. S5A, B). Silencing of TFEB by siRNA significantly decreased the fraction of EGFP-FYVE that was found on LAMP1-positve lysosomes in JMSU1 cells (Fig. 5B). Moreover, measuring the global cellular level of EGFP-FYVE showed a significant reduction on endomembranes after knock down of TFEB (Fig. 5C, D).

The majority of PtdIns3P is produced by class III PI3 kinase (PIK3C3/VPS34), which converts phosphatidylinositol (PI) to phosphatidylinositol-3-phosphate[29]. We therefore further investigated whether TFEB could regulate protein expression of PIK3C3. We found that siRNA-mediated TFEB depletion significantly decreased protein and mRNA levels of PIK3C3 (Fig. 5E, F and Fig. S5C).

To further validate the TFEB-dependent upregulation of endosomal PtdIns3P and recruitment of PtdIns3P-binding proteins to endosomal membranes we analyzed EEA1, a well-studied FYVE containing protein. We found a significant increase of EEA1 on endomembranes upon treatment of RT112 cells with rapamycin or starvation (Fig. S5D, E). Interestingly and different to protrudin, this upregulation was dependent on TFEB, because knock down of TFEB prior to rapamycin treatment abolished the increase of EEA1 on endomembranes (Fig. S5F, G). Basal level of EEA1 recruitment on endosomes was also decreased in RT112 upon siTFEB (without any additional treatment) (Fig. S5F, G). Note that EEA1 recruitment was also dependent on protein translation, because no increase of endosomal EEA1 was observed in the first 4 h after rapamycin treatment (Fig. S5D, E) and in the presence of the protein translation inhibitor cycloheximide along with rapamycin treatment (Fig. S5D, E). Total EEA1 protein levels did not majorly change under all tested conditions (Fig. S5H, I), although EEA1 expression has been previously described to be under the control of TFEB[30]. Conversely, gene silencing of TFEB in JMSU1 cells significantly decreased EEA1 levels on endosomes without affecting the total amount of EEA1 protein level (Fig. S5J–L) indicating a decrease of PtdIns3P in the endosomal pathway after TFEB depletion.

Finally, we tested the role of endosomal PtdIns3P levels on protein recruitment and lysosome positioning. As expected, inhibition of phosphatidylinositol-3-phosphate kinases by wortmannin in high-grade JMSU1 cells strongly depleted

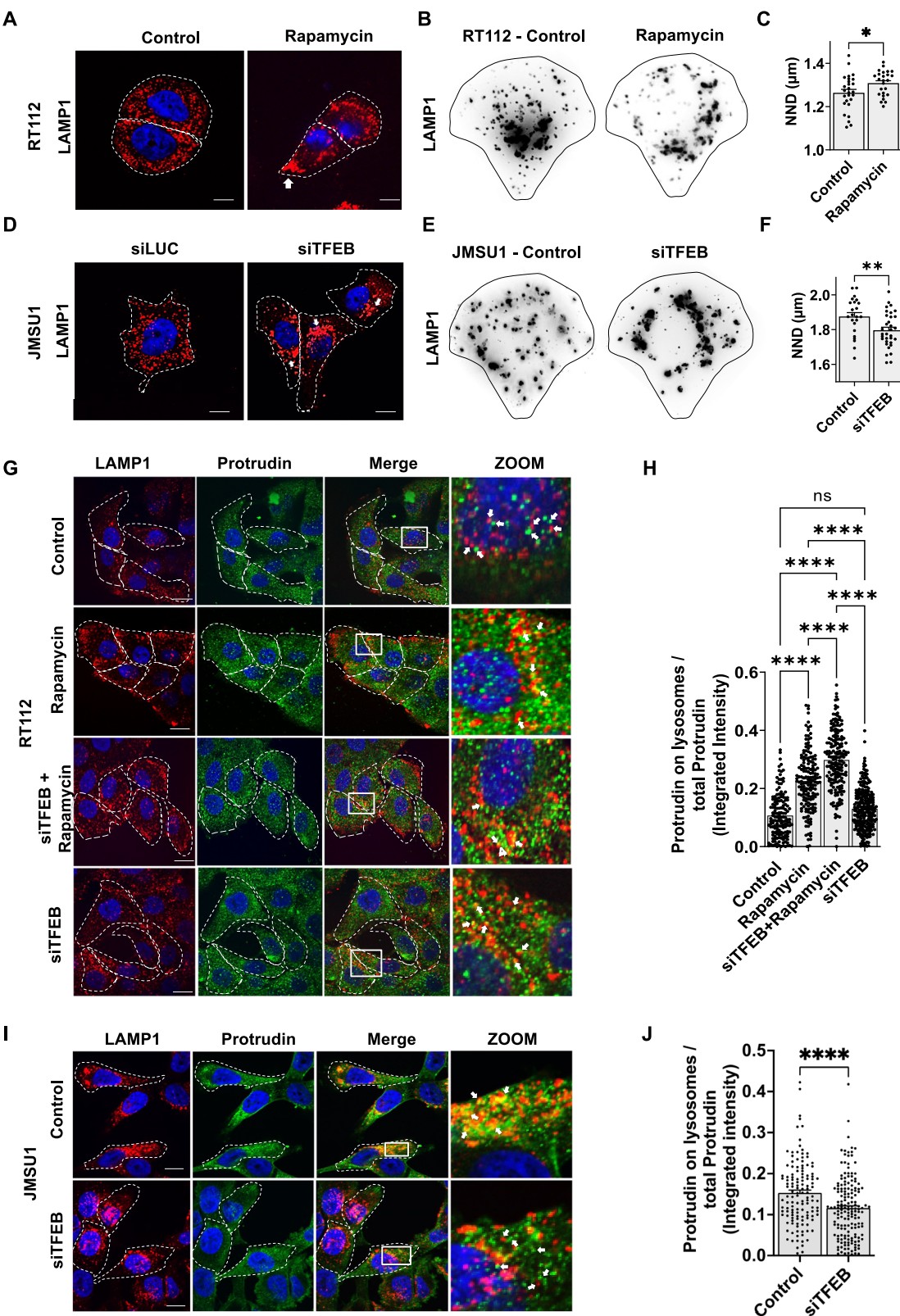

EEA1 from endosomes (Fig. S5M, N), mimicking the phenotype of TFEB knock down (Fig. S5J, K). Moreover, wortmannin treatment induced the central clustering of lysosomes in JMSU1 cells, leading to a significant reduction of the NND of lysosomes (Fig. 5G–I). This showed that dispersion of lysosomes towards cell periphery requires endosomal PtdIns3P. Altogether, our results indicate that endosomal PtdIns3P levels dictate lysosomal positioning and are regulated by TFEB in bladder cancer cells.

## Discussion

Our study identifies and characterizes a remarkable cellular phenotype of aggressive malignancy in a model of bladder cancer.

**Fig. 4 Lysosome positioning changes are under the control of TFEB in JMSU1 cells. A** Immunofluorescence staining of the lysosomal-associated membrane protein 1 (LAMP1, CD107a) in control (DMSO) and rapamycin (10 µM) treated RT112 cells. White arrow shows the peripheral clustering of lysosomes. Scale bars equal 10 µm. **B** Representative images of lysosomes visualized by immunofluorescence staining against LAMP1 in micropatterned RT112 cells in control and rapamycin treatment. **C** Nearest neighbor distance (NND in µm) between lysosomes in micropatterned control ($n = 25$) and rapamycin treated ($n = 27$) RT112 cells; *$p < 0.05$ in a Mann–Whitney U test, error bars are SEM. **D** Immunofluorescence staining of LAMP1 in JMSU1 cells treated with siLUC and siTFEB for 72 h. White arrow shows the perinuclear clustering of lysosomes. Scale bars equal 10 µm. **E** Representative images of lysosomes visualized by immunofluorescence staining against LAMP1 in micropatterned JMSU1 cells in control (siLUC) and siTFEB treatment conditions. **F** Nearest neighbor distance (NND in µm) between lysosomes in micropatterned control (siLUC) ($n = 23$) and siTFEB ($n = 33$) treated JMSU1 cells; **$p < 0.005$ in a Mann–Whitney U test, error bars are SEM. **G** Immunofluorescence staining of LAMP1 (red) and protrudin (green) in control (siLUC 72 h + DMSO 24 h), rapamycin (10 µM, 24 h), siTFEB (72 h) + rapamycin (10 µM, 24 h) and siTFEB (72 h) treated RT112 cells. Zoom shows the merged image of both proteins in the white box. White arrow shows the colocalization between LAMP1 and protrudin. Scale bars are 15 µm. **H** Quantification of protrudin integrated intensity on lysosomes normalized to total cellular protrudin, in 154 control, 167 rapamycin (24 h), 184 siTFEB (72 h) + rapamycin (24 h) and 242 siTFEB (72 h) treated RT112 cells; ns $p > 0.05$, ****$p < 0.0001$ in a Kruskal-Wallis test with Dunn's test for multiple comparison, error bars are SEM. **I** Immunofluorescence staining of LAMP1 (red) in and protrudin (green) in JMSU1 cells in control (siLUC) and siTFEB (72 h) treatment conditions. Zoom shows the merged image of the two proteins in the white box. White arrow shows the colocalization between LAMP1 and protrudin. Scale bars are 15 µm. **J** Quantification of protrudin integrated intensity on lysosomes normalized to total cellular protrudin, in 131 control (siLUC) and 167 siTFEB JMSU1 cells; ****$p < 0.0001$ in a Mann–Whitney U test, error bars are SEM.

We show that the lysosomal compartment is scattered to the cell periphery in all analyzed cell lines representing high-grade bladder cancers, a phenotype that we did not see in normal urothelial cells. Lysosomal dispersion or scattering has been reported in cancers such as breast cancer[31], prostate cancer[32] or hepatocellular carcinomas[33], where this phenotype has been shown to be associated with increased cancer invasiveness. The phenotype of lysosomal dispersion is different to the previously described expansion of the lysosomal compartment in pancreatic ductal adenocarcinoma (PDA) and indicative of increased lysosome biogenesis[34]. Indeed, we did not observe a systematic increase in lysosome numbers or size in cell lines representing high-grade cancers.

We show that lysosome positioning changes are correlated with changes in the mTORC1-TFEB signaling axis. Firstly, we find the mTORC1 substrate 4EBP1 is highly expressed and phosphorylated in JMSU1 cells, whereas another substrate, p70-S6K1, is highly phosphorylated in MGHU3 and RT112 cells. Previous studies in bladder cancer have shown that overexpression of 4EBP1 correlated with increased infiltration of cancer associated fibroblasts (CAFs) and resulted in poor prognosis[35]. Besides these alterations in phosphorylation of different substrates, our results indicate that mTORC1 is active in cell lines with peripheral as well as central lysosomes. Korolchuk et al. have shown that activation of mTORC1 by nutrients correlates with its presence on peripheral lysosomes[18]. Our results reveal an additional complexity, showing that lysosome positioning could potentially correlate with differential substrate phosphorylation, in addition to mTORC1 activity. Further studies will be required to fully comprehend mTORC1 regulation by lysosome positioning. Secondly, we found that TFEB nuclear translocation was induced by rapamycin in RT112 cells. TFEB has been shown to be a rapamycin-insensitive mTORC1 substrate in several seminal studies[8,36,37]. Other studies have shown that rapamycin can cause TFEB nuclear translocation in cells[38,39], which could be due to rapamycin induced activation of mucolipin-1 (TRPML1)[24] and subsequent activation of calcineurin[25]. However, inhibiting mucolipin-1 by ML-SI1 did not prevent nuclear translocation of TFEB upon rapamycin treatment in RT112 cells. Thus, it is not clear by which molecular mechanisms rapamycin-dependent nuclear translocation of TFEB is sustained. Lastly, besides active mTORC1, we observed that TFEB was translocated to the nucleus in KU19-19 and JMSU1 cells. Because, inhibition of mucolipin-1 (TRPML1) or chelating Ca$^{2+}$ prevented nuclear translocation of TFEB in JMSU1 cells, TFEB seems to be under the control of the mucolipin-1/ calcineurin axis in these cells. Together our results indicate that the calcium/ mucolipin-1/ calcineurin axis seems to

be an important target to study the lysosomal phenotype in bladder cancer. Deregulation of mTORC1-TFEB regulatory circuit parallels previous studies in other cancer types[3,4,10,40,41] and aligns with a genetic study of NMIBC that identified alterations in mTORC1 signaling in several bladder cancer subtypes[42].

Peripheral dispersion of lysosomes has been previously reported in prostate cancer cells due to the acidification of the extracellular milieu[43]. Such a mechanism is unlikely in the case of bladder cancer cells, because all cells used in this study were grown in the same pH-buffering medium. Investigating lysosome-related mechanisms, we found that lysosome positioning is controlled by PtdIns3P in bladder cancer cell lines. Increased levels of PtdIns3P on endosomes leads to enhanced recruitment of FYVE-containing proteins such as protrudin[27]. Protrudin is found at membrane contact sites between lysosomes and the endoplasmic reticulum (ER) and recruits the kinesin-1 adaptors FYCO1 to promote the microtubule-dependent translocation of lysosomes to the cell periphery[44,45]. However, several alternative pathways for anterograde lysosome trafficking have been described that all require endosomal PtdIns3P and could additionally be harnessed by cancer cells. The alternative kinesin-1 adaptor SKIP (also known as PLEKHM2) also contains three lipid-binding pleckstrin homology (PH) that conceivably could bind to lysosomal PtdIns3P. Moreover, KIF16B, a highly processive kinesin-3 family member that participates in the trafficking of endosomes along microtubules contains a PX (Phox homology) motif binding PtdIns3P[46]. PtdIns3P formation depends mainly on the class III PI3 kinase (PIK3C3/ Vps34)[29]. Notably, VPS34 is recruited by active mucolipin-1 (TRPML1) via a cascade including calcium/calmodulin-dependent protein kinase kinase β (CaMKKβ), AMP-activated protein kinase (AMPK) and ULK1 kinase[47]. Because rapamycin directly activates mucolipin-1[24], PtdIns3P production via this pathway could be sufficient for the binding of protrudin[27] in RT112 cells and to induce anterograde lysosome movement. Yet, our data indicate that PtdIns3P production is also transcriptionally regulated and under the control of TFEB. We show that TFEB depletion in JMSU1 cells decreased PIK3C3 (VPS34) mRNA and protein levels as well as the binding of FYVE-domain-containing proteins or the PtdIns3P probe EGFP-FYVE, indicating endosomal PtdIns3P loss. Interestingly, rapamycin-induced EEA1 recruitment on endosomes in RT112 was also TFEB- dependent, in contrast to protrudin. Support for transcriptional regulation of EEA1 recruitment in these cells comes also from the fact that endosomal EEA1 was not increased in the first 4 h after rapamycin treatment but only after 24 h, and was inhibited by cycloheximide. Thus, several parallel mechanisms seem to

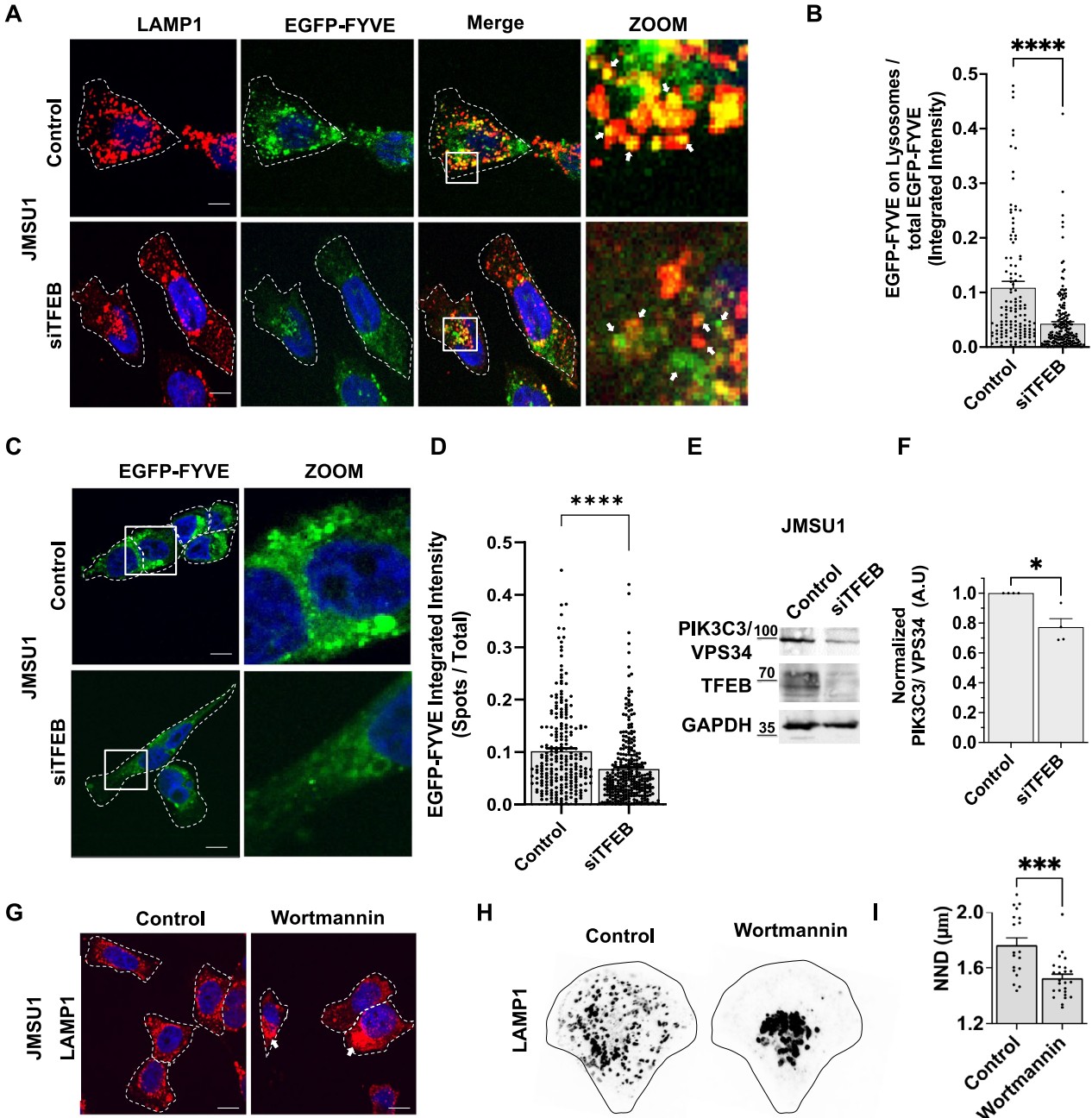

**Fig. 5 TFEB regulates phosphatidylinositol-3-phosphate levels on endomembranes in bladder cancer cells. A** Immunofluorescence staining of the lysosomal-associated membrane protein 1 (LAMP1/CD107a) (red) in control (siLUC) and siTFEB (72 h) treated JMSU1 cells transfected with EGFP-FYVE (green). Zoom shows the merged images of LAMP1 and EGFP-FYVE in white box. White arrow shows the colocalization between LAMP1 and EGFP-FYVE. Scale bars equal 10 µm. **B** Quantification of EGFP-FYVE integrated intensity on lysosomes normalized to total cellular EGFP-FYVE, in 132 siLUC and 188 siTFEB treated JMSU1 cells; ****$p < 0.0001$ in a Mann–Whitney U test, error bars are SEM. **C** Representative images of control (siLUC) and siTFEB (72 h) treated JMSU1 cells expressing EGFP-FYVE. Zoom shows EGFP-FYVE in white box. Scale bars equal 15 µm. **D** Quantification of EGFP-FYVE integrated intensity on segmented spots normalized to corresponding total cellular EGFP-FYVE, in 176 siLUC and 244 siTFEB JMSU1 cells; ****$p < 0.0001$ in a Mann–Whitney U test, error bars are SEM. **E** Western blot analysis of PIK3C3/ VPS34 in JMSU1 cells in control (siLUC) and siTFEB (72 h) treatment conditions. **F** Western blot quantification of PIK3C3/ VPS34 in JMSU1 cells in control (siLUC) and siTFEB (72 h) treatment conditions from $n = 4$ experiments. *$p < 0.05$ in a $t$ test, Error bars are SD. **G** Immunofluorescence staining of LAMP1 in control (DMSO) and wortmannin (1 µM) treated JMSU1 cells. White arrows show the perinuclear clustering of lysosomes. **H** Representative images of lysosomes visualized by immunofluorescence staining against LAMP1 in micropatterned control and wortmannin (1 µM) treated JMSU1 cells. **I** Nearest neighbor distance (NND in µm) between lysosomes in micropatterned in control ($n = 19$) and wortmannin ($n = 25$) JMSU1 cells; ***$p < 0.001$ $p$ value in a Mann–Whitney U test, error bars are SEM.

regulate PtdIns3P production in bladder cancer cell lines that result in TFEB-dependent and independent lysosome movements. Other studies have reported that TFEB regulated lysosome positioning[26,48]. For instance, activation of TFEB in HeLa cells

has been shown to upregulate the lysosomal transmembrane protein TMEM55B resulting in central clustering of lysosomes through its interaction with dynein adapter JIP4[26]. On the other hand, TFEB overexpression has been also shown to regulate

lysosome docking to plasma membrane and lysosomal exocytosis by upregulating $Ca^{+2}$ levels through activation of mucolipin-1 (TRPML1)[48,49]. This complexity in the regulation of lysosome positioning by several overlapping as well as alternative pathways found in (cancer) cells could contribute to the discrepancies between studies. It seems that mechanisms/pathway are predominant in different cells and additionally may act at several time scales as part of various feedback regulations. Thus, targeting TFEB or using potent inhibitors such as rapamycin that interfere with several players of the mTORC1-TFEB axis could converge on opposite phenotypes on lysosome positioning[18,26,48]. Our results indicate that in bladder cancer cell lines mucolipin-1-dependent mechanisms are more predominant than TMEM55B-dependent ones. Together our results propose that the lysosome phenotype in bladder cancer cells is controlled by endosomal PtdIns3P production by VPS34 that are partly under transcriptional regulation by TFEB. VPS34 has been proposed to be regulated by TFEB recently[50], however it is not found in the CLEAR genes list[22,23]. Therefore, further studies are needed to shed light on the molecular mechanisms of regulation. Intriguingly, in addition to PtdIns3P, the ubiquitin system has been shown to regulate lysosome positioning, particularly the ER-embedded UBE2J1/RNF26 ubiquitylation complex and the opposing deubiquitinating enzyme USP15[51,52], the ubiquitin domain-containing protein 1 (UBTD1)[53] and deubiquitinase USP17[54]. Further investigations will reveal whether these represent an alternative pathway or whether ubiquitylation cross talks with phosphoinositide regulation.

Interestingly, it has been shown that endosomal PtdIns3P levels regulate mTORC1 recruitment and signaling via amino acids and stimulation of PIK3C3 (VPS34)-mediated PtdIns3P synthesis[55]. PtdIns3P also facilitates lysosomal recruitment of phospholipase D1 (PLD1), which triggers dissociation of the inhibitory DEP-TOR subunit from mTORC1[56]. Formation of phosphatidylinositol 3,5-bisphosphate (PtdIns3,5P$_2$) from PtdIns3P additionally regulates mTORC1 via raptor[57]. Contrary, the PtdIns3P3-phosphatase MTMR3 interacts with mTORC1, and over-expression of this enzyme inhibits mTORC1 activity[58]. As increased endosomal PtdIns3P levels globally activate mTORC1 that deactivates TFEB, we speculate that PtdIns3P could be part of a feedback of TFEB on mTORC1[30]. Our current understanding is that nutrient status, pH and growth factors assemble a sophisticated machinery on the surface of lysosomes to integrate the different inputs upstream of mTORC1[1,2,59]. Because PtdIns3P and several motor proteins/adapters are part of this machinery, mTORC1 signaling is coupled with lysosome positioning[18]. Our data are consistent with the following model: In bladder cell lines representing high-grade cancers, TFEB localization is mostly nuclear and active thus leading to an increase in PtdIns3P levels on different endomembranes, including lysosomes. This increase leads to the recruitment of FYVE-domain containing proteins such as EEA1 and protrudin and supports anterograde movement of lysosomes. The anterograde movement gives rise to the typical signature of peripheral lysosomes that we find in bladder cancer cell lines. Peripheral lysosomes have been shown to recruit more mTORC1 and increase phosphorylation of downstream substrates[3,18,27]. This would allow a feedback control of TFEB by mTORC1. However, this seems not to occur in bladder cell lines representing high-grade cancers, because mTOR levels on lysosomes do not increase. Instead, the efficient calcium-dependent dephosphorylation of TFEB hinders its cytoplasmic translocation and control by mTORC1. Together, our results provide a mechanistic explanation to the characteristic cellular phenotype of lysosome dispersion in cell lines representing high-grade bladder cancers. Yet, further studies will be required to reveal in detail the deregulation of the mTORC1/TFEB axis in these cells.

In addition to signaling, lysosome positioning has been implicated in the regulation of protease secretion/proteolysis[44], migration[60–63] and remodeling of the tumor environment through the release of exosomes[64]. Thus, it is tempting to speculate that altered lysosome signaling could link dysfunctional cancer cell metabolism with cancer invasiveness.

In conclusion, we discovered unexpected phenotypes in terms of TFEB dynamics in aggressive bladder cancer cells lines. Future studies will be required to comprehensively depict the mechanisms regulating TFEB as well as their relevance in patients and disease progression. In addition to revealing a distinct cellular phenotype characteristic of cancer cells together with the underlying molecular mechanism, our results uncover an unexpected role of TFEB in regulating PtdIns3Ps levels on endosomes. Several studies have illustrated the crucial role of TFEB in regulating fundamental but distinct cellular processes such as endocytosis, lysosomal biogenesis and autophagy. Because these different compartments of the endolysosomal system retain their identities based on the lipid composition of their membranes and are regulated by PtdIns3P levels, our results conceptually clarify the role of TFEB as regulator of endosomal maturation.

## Material and methods

**Cell culture and treatments**. Bladder cancer cells lines RT4, MGHU3, RT112, KU19-19, JMSU1, T24 and TCCSup were grown in RPMI-1640 medium (Life Technologies, Carlsbad, CA, USA), supplemented with 10% Fetal Bovine Serum (FBS; Eurobio, Courtaboeuf, France). Normal human urothelium (NHU) cells were from Jennifer Southgate (University of York, UK). NHU were grown in KSFMC medium[65]. For experiments with inhibitors, as per the experiment either the day after cell seeding or after transfection respective drugs were added for incubation time of 24 h or as indicated and cells were incubated, at 37 °C. The concentration of inhibitors used were as follows: rapamycin (10 μM), wortmannin (1 μM, 2 h), ML-SI1 (20 μM, 3 h), BAPTA AM (10 μM, 3 h) and cycloheximide (20 μg/mL). For starvation experiments, the day after cell seeding, the medium was removed and cells were washed once with EBSS (Earle's Balanced Salt Solution) and incubated in EBSS for 4 or 24 h, as per the experiment, before lysate preparation or cell fixation with 4% PFA.

**Cell transfection**. For RNA interference studies, 200,000 cells were transfected in 12 well plate with 25 pmol siRNA (siTFEB: ON-TARGETplus Human TFEB, L-009798-00-0005, Dharmacon™) using Lipofectamine RNAiMAX Transfection Reagent (1:200; Life Technologies). Cells were incubated 72 h at 37 °C prior to further manipulation or drug treatment. Efficiency of gene silencing was verified by western blot of cell lysate after three days of transfection.

For plasmid transfection, 200,000 cells were transfected in a 12 well plate. Transfection was performed using Lipofectamine LTX with Plus reagent (Invitrogen) using 1 μg of plasmid. pEGFP-N1-TFEB plasmid was a gift from Shawn Ferguson (Addgene plasmid # 38119; http://n2t.net/addgene:38119; RRID:Addgene_38119n[36]) or EGFP-2X FYVE plasmid (kind gift from B. Payrastre, Toulouse). 48 h post transfection, cells were trypsinized and transferred to sterilized coverslips (12 mm) in 1 mL medium in 12 well plate. Cells were fixed with 4%PFA 72 h after transfection and used for immunofluorescence and imaging.

**Micro-array analysis**. Micro array data were analyzed with R (3.5.2). The annotation was performed using affy package (1.58.0) with a custom CDF (Chip Description File) from brain array (huex10st, genome version 23). Normalization was done with

RMA algorithm using affy library[66] and batch effect corrected with ComBat[67]. The PCA was computed from these normalized and corrected data.

**qRT-PCR.** Total RNA was isolated from JMSU1 cells using Trizol reagent (Life technologies). cDNA was prepared using one microgram total RNA with high capacity cDNA reverse transcriptase kit and the manufacturer's protocol (Life technologies). Quantitative PCR was performed using KAPA SYBR FAST (Sigma-Aldrich) and the following primers for the PIK3C3 gene: forward, 5′-GACAGGCCGATGATGAGGATTT-3′ and reverse 5′-AAGGGGGCTGGTTATAATTTGGG-3′. To normalize the expression beta actin gene was used as normalizing control with the following primers: forward 5′-CATGTACGTTGCTATCCAGGC-3′ and reverse 5′-CTCCTTAATGTCACGCACGAT-3′. Samples were run in triplicate and gene of interest expression was normalized to human beta actin.

**Micropatterned coverslips preparation and cell seeding.** Micropattern production was performed using photo-lithography methods[16,17]. Briefly, coverslips were coated with Poly-L-Lysine(20)-grafted[3.5]-Polyethyleneglycol(2) (PLL-g-PEG) from SuSoS (Dübendorf, Switzerland) at a final concentration of 0.1 mg/mL in 10 mM HEPES (pH 7,3) solution. Coverslips were exposed to deep UV during 5 min using a photomask containing arrays of crossbows (37 μm diameter, 7 μm thick). Prior to cell seeding, the patterned surface was incubated for 1 h with a mixture of 50 μg/mL fibronectin (Sigma-Aldrich, St. Louis, MO, USA), 5 μg/mL concanavalin A (Sigma-Aldrich, St. Louis, MO, USA) and 1 μg/mL fibrinogen–Cy5 (Invitrogen). Cells were seeded on micropatterns in RPMI medium supplemented with 20 mM HEPES (Life Technologies) for 4 h prior the experiment.

**Invasion assay.** Cells were trypsinized and $10^4$ cells/ml were resuspended in RPMI medium containing 10% FBS and 1% Penicillin-Streptomycin (Life Technologies). Then 100 μl of cell suspension was plated in 48-well plates coated with 1% agarose (Life Technologies) and incubated for 3 days. In each well, a spheroid was formed from $10^3$ cells. Next, the spheroids were plated on Lab-Tek chambers (Sigma), in a mixture of collagen I from rat tail (Corning) at a final concentration of 2 mg/ml, PBS, sodium hydroxide (NaOH) and serum-free medium. The spheroids were monitored for 5 consecutive days by using an inverted Leica microscope (Wetzlar, Alemanha) equipped with camera device using 4x objective.

**Immunofluorescence.** Cells were fixed with 4% formaldehyde for 15 min at room temperature, washed three times with PBS and permeabilized in PBS containing 0.2% BSA and 0.05% saponin. Cells were then incubated with the primary antibodies in PBS containing 0.2% BSA and 0.05% saponin (anti-Lamp1/CD107a, 555798, BP Pharmingen™, 1:1000; anti-mTOR, 7C10, #2983, Cell Signaling Technology, 1:1000; anti-EEA1, 610456, BD Biosciences, 1:500; protrudin/ ZFYVE27, 12680-1-AP, Proteintech, 1:500) and Alexa Fluor 488, Alexa Fluor 647 or Cy3-coupled secondary antibodies (Jackson ImmunoResearch, 1:400 in PBS containing 0.2% BSA and 0.05% saponin) for 1 h. Actin was visualized by FluoProbes 547H (557/572 nm) coupled Phalloïdin (Interchim) and nuclei with 0.2 μg/mL 4',6-diamidino-2-phenylindole (DAPI; Sigma-Aldrich). Coverslips were mounted in Mowiol (Sigma-Aldrich).

**Western blot.** In total, 250,000 cells were seeded in a 12 well plate one day prior to the experiment. Drug treatments or knock-down experiments were performed as mentioned before. Equal volumes of lysate from each cell line was loaded on a 10% or 12% polyacrylamide gel, resolved by SDS-PAGE and electrotransferred to nitrocellulose membranes. Membranes were incubated with primary antibodies at 4 °C overnight: Phospho P-70 (Thr389)-S6K (CST: 9205 S, 1:1000 in 5% BSA in TBST), P-70 S6K (CST: 9202 S, 1:1000 in 5% milk in TBST), Phospho (Ser65)-4EBP1 (CST: 9451, 1:1000 in 5% BSA in TBST), 4EBP1(CST: 9452, 1:1000 in 5% milk in TBST), GAPDH (Sigma: G9545, 1:10,000 in 5% milk in TBST), EEA1(610456, BD Biosciences, 1:500 in 5% milk in TBST), protrudin (ZFYVE27, Proteintech 12680-1-AP), TFEB (CST: 4240, 1:1000 in 5% milk in TBST), Lamin B1 (Abcam: ab16048, 1:500 in 5% milk in TBST) and species specific HRP secondary antibodies (1:10,000) for 1 hour at room temperature, following ECL western blotting substrate.

**Image acquisition.** Images for immunolabelled cells on micropatterns were acquired with an inverted wide field Deltavision Core Microscope (Applied Precision) equipped with highly sensitive cooled interlined charge-coupled device (CCD) camera (CoolSnap Hq2, Photometrics). Z-dimension series were acquired every 0.5 μm.

Images for non-patterned immunolabelled cells were acquired with a spinning disk confocal microscope (Inverted Eclipse Ti-E (Nikon) + spinning disk CSU-X1 (Yokogawa) integrated with Metamorph software by Gataca Systems). Cells were imaged as Z-stacks with 0.2 μm distance and 12 μm total height.

**Image processing and analysis.** For cells on micropatterns, several tens of single cell images were aligned using the coordinates of the micropattern[17,68] determined via an ImageJ-based macro (Bethesda, MD, USA). To extract the 3D spatial coordinates of lysosomes, images were segmented with the multidimensional image analysis (MIA) interface on MetaMorph (Molecular Devices, Sunnyvale, CA, USA) based on wavelet decomposition. The coordinates of the segmented structures were processed for density estimation[17] programmed in the ks library in R. For visualizing kernel density estimates, probability contours were visualized using the extension libraries mvtnorm, rgl, and miscd. The lysosome volume was estimated using the sum of all pixels in all 2D planes of one segmented lysosome (Z-planes were acquired every 0.5 μm). This value has been obtained from the MIA segmentation interface on MetaMorph (Molecular Devices, Sunnyvale, CA, USA) used in batch on all cells and represents arbitrary units (a.u.) of lysosome volume measurement.

Levels of lysosome dispersion in non patterned MGHU3, RT112, KU19-19 and JMSU1 cells were measured using statistical inertia (=averaged squared distance to the center of mass). To control for variations in cell size differences, normalization to cell size has been applied. Lysosome coordinates have been divided by the coordinates of the center of the mass (setting the center mass at x = 1, y = 1). This quantifies the dispersing of the lysosome structures independently of homogeneous dilations due to cell size. To test statistical significance, a Kruskal-Wallis test with Dunn post-hoc test with Sidak correction for multiple comparisons has been applied.

Image analysis for the Figs. (2B, 4H, J, 5B, D, and S4B, G, S5B, E, G, K, N) was done using CellProfiler (version: 3.1.9) on one Z-plane of the images. The pipelines for different analysis were prepared as follows:

*To detect the total and membrane bound intensities of protein of interest (labeled as total integrated intensity or spots/total, respectively, in the figures) or intensities of co-localized proteins the pipeline was created as follows:*

Step 1: Module 'EnhanceorSuppressFeatures' was applied to channels where the objects needs to be segmented, either to

obtain their intensities or objects for the intensities of co-localized proteins, to get sharp and defined objects which makes segmentation easier (for eg. On channels with LAMP1 or EEA1 or GFP-FYVE).

Step 2: Nucleus was identified in the DAPI channel using the 'IndentifyPrimaryObject' module

Step 3: Module 'IndentifyPrimaryObject' was used again on the images obtained from Step 1 to segments objects whose measurements are required (such LAMP1, EEA1, EGFP-FYVE)

Step 4: Cells were segmented using the 'IdentifySecondaryObject' module with nucleus as the 'primary object' (identified in step 2) and using phalloidin or another cytoplasmic protein channel to recognize the cell boundaries.

Step 5: Module 'RelateObjects' was used to relate the objects obtained in step 3 to each cell obtained in Step 4. Output of this channel was saved as another object which gives the objects of protein of interest per cell.

Step 6: Objects from step 3 were masked on the channel whose co-location or membrane bound fraction had to be calculated using the 'MaskImage' module (for e.g.: to calculate EGFP-FYVE on lysosomes in Fig. 5B, Lysosomes were segmented in step 3 and the output objects were masked on EGFP-FYVE channel or to calculate membrane bound EGFP-FYVE, segmentation of EGFP-FYVE objects from step 3 was masked on EGFP-FYVE channel). Output of this step was saved as a new image in the pipeline.

Step 7: 'MeasureObjectIntensity' module was used to obtain total 'per cell intensity' and 'intensity on spots' of protein of interest. Intensities were picked from images from step 6 and raw images of channel of interest using cells from step 4 as the objects.

Step 8: Cell size was obtained using the module 'MeasureObjectSizeandShape' on the cells segments in Step 4 as the objects.

Step 9: Finally, all the measurements were exported to the excel sheet using the module 'ExporttoSpreadsheet'.

Step 10: The final values were exported to a csv file named 'cell'. This file had the values of cell size (in pixels), total intensity of protein of interest per cell, intensity of protein of interest on spots and intensity of co-localized protein on the object of interest (e.g.: GFP-FYVE on lysosomes).

Integrated intensities were used for the analysis and to plot the graphs. For measurements of EEA1 or EGFP-FYVE intensities on endosomes (Fig. 5D; S5D–G; S5J–N) the 'intensity on spots' was normalized to the total fluorescence ('per cell intensity') of the cell analyzed.

**Cell fragmentation**. 1.5 million cells were seeded in a 10 cm dish 2-3 days before the experiment and allowed to grow. After 3 days, cells were washed once in ice cold PBS, pH 7.4 and scrapped using a plastic scrapper in 1.5 mL ice cold PBS. Cells were centrifuged for 10 sec in a tabletop pop-spin centrifuge and the supernatant were discarded. The pellet was resuspended in 1.2 mL of ice cold 0.1% NP40 (Sigma Aldrich) in PBS. 400 µL of mix was removed as the whole cell lysate, 100 µL of 5X Laemmli buffer was added, and the sample was boiled for 1 min. The rest of the sample was centrifuged as above. 400 µL of supernatant was collected as the cytosolic fraction, 100 µL of 5X Laemmli buffer was added, and the sample was boiled for 1 min. The rest of the supernatant was discarded. The pellet containing the nuclear fraction was resuspended in 400 µL of 1X Laemmli buffer, and the sample was boiled for 1 min. The protocol was adapted from Suzuki et al., 2010[69].

**GSEA analysis**. GSEA has been made using fgsea() function from fgsea package. High-grade condition agglomerates 2 TCCSup, 2

T24, 1 JMSU1 and 1 KU1919 transcriptomic data, and low-grade agglomerates 1 RT4, 3 MGHU3 and 2 RT112 transcriptomic data.

**Statistics and reproducibility**. The statistical analysis of endo-lysosome volume, number and normalized NND was performed with R (3.6.0). For NND analysis, the centroids distance between structures was calculated from a constant number of lysosomes that was randomly sampled from each cell. Therefore, variation in NNDs cannot be imputed to variation in the number of lysosomes but to bona-fide variation of their spatial organization. The statistical analysis was a Kruskal-Wallis test with Dunn test for multiple comparisons correction.

For all experiments, a large number of cells were monitored from 3 to 6 independent experiments. Two-sided Mann-Whitney U test were performed for 2 conditions comparisons. For multiple comparisons, a Kruskal-Wallis has been used with Dunn's test for multiple comparisons. Additionally, to compare the global distribution of cell population, $\chi^2$ tests were performed (R function "chi-square()") and Benjamini-Hochberg multiple comparison correction has been applied. For the statistical analysis on the data from CellProfiler, GraphPad Prism 9.3.1 was used. Mann-Whitney U test or paired t test was applied for the two conditions comparison (indicated in respective figures) or Kruskall-Wallis test with Dunn test for multiple comparison.

**Reporting summary**. Further information on research design is available in the Nature Portfolio Reporting Summary linked to this article.

## Data availability

All data used for plotting of the main figures can be obtained from the "Supplementary-data-Fig" 1–5 and the legend of these data can be found in the "Supplementary data legends" provided with the manuscript. The original, uncropped WB images are shown in Fig. S6. All other data are available from the corresponding author upon request.

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

## Acknowledgements

We thank Jennifer Southgate (University of York, UK) for her gift of NHU cell extracts and Clémentine Krucker and Yann Neuzillet (Foch Hospital, France) for deriving these cells. We thank Elodie Chapeaublanc for help on the bioinformatics and Oliver Kepp for critical reading of the manuscript. The authors greatly acknowledge the Cell and Tissue Imaging Facility (PICT-IBiSA @Burg and @Pasteur) and the Nikon Imaging Center at Institut Curie (Paris) that are member of the French National Research Infrastructure France-BioImaging (ANR10-INBS-04). This project was supported by grants from the European Union's Horizon 2020 research and innovation program under the Marie Skłodowska-Curie grant agreement n° 666003 to PM; Capes/ Ciência sem Fronteiras/Process (9121137) to CDBS; Grants from Fondation ARC pour la recherche sur le cancer to K.S. and H.L., Institut Curie SIRIC grant to KS, Agence Nationale de la Recherche (#2010 BLAN 122902), the ITMO Nanotumor grant to KS, the Center National de la Recherche Scientifique and Institut Curie. The Goud team is member of the Labex Cell(n)Scale (11-LBX-0038) and the Idex Paris Sciences et Lettres (ANR-10-IDEX-0001-02 PSL). The Molecular Oncology team (FR) is supported by La Ligue Contre le Cancer (Equipe labellisée program).

## Author contributions

P.M. contributed to the design of the study, planned and conducted the experiments, prepared all figures and wrote the manuscript. C.D.B.C. planned and conducted the experiments of Fig. 1 and S1, H.L. performed transcriptome analysis and NND analysis, J.P. performed the experiments and analyses of Fig. S5C, B.L. performed analyses for Fig. 1 and S1, F.R. provided bladder cancer cell lines, NHU cells and the transcriptome data for cancer cell lines, B.G. contributed to the study design, K.S. designed, drafter and conceptualized the study and wrote the manuscript.

## Competing interests

The authors declare no competing interests.
