## [Peer Review File · Communications Biology]

Reviewers' comments:

Reviewer #1 (Remarks to the Author):

The manuscript reports a distinctive lysosome phenotype of bladder cancer cells consisting in peripheral lysosome positioning. This phenotype is accompanied by downregulated mTOR signalling and nuclear translocation of overexpressed GFP-TFEB. TFEB silencing reverts the lysosome scattering and also reduces the amount of LAMP1-associated protrudin. The authors also report that bladder cancer cells exhibit higher endosome association of the transiently overexpressed PI3P probe EGFP-FYVE domain and that this increased association is mimicked by TFEB overexpression and blunted by TFEB depletion.

While the authors identify a distinctive phenotype that could be used "as a potential biomarker for bladder cancer" they do not provide mechanistic insights linking TFEB activity to lysosome positioning and to PI3P production.

Specific comments

- The data showing TFEB activation in high-grade bladder cancer cell lines have been performed upon TFEB overexpression only. The analysis of endogenous TFEB in these cells is needed as TFEB overexpression may lead to artifacts. For instance, TFEB is a well-known rapamycin-insensitive mTORC1 substrate (PMID: 21617040, 22343943, 22692423). Thus, the data showing that rapamycin treatment results in TFEB nuclear translocation (Figure 2) are puzzling and are in contrast with previously published and well-established results.

- Rapamycin can activate TRPML1 in an mTOR independent manner (PMID: 31112550). Thus, the possibility exists that in using rapamycin the authors are activating TRPML1. Indeed, from the experimental evidence presented, it looks as though mTORC1 is weakly affected, and therefore a role of calcium-TRPML1-Ca²⁺ might represent a better target to study this lysosomal phenotype. TRPML1 was reported to regulate PI3P production (PMID: 31822666). This observation together with the author's data might suggest that this pathway may play a major role in TFEB regulation in the bladder cancer cells.

- Furthermore, the data in Supplementary Figure 2C show a very mild upregulation of two TFEB-target genes (RRAGD and TSC1), indicating little or no activation of endogenous TFEB in bladder cancer cell lines.

The authors performed transcriptome analyses in several bladder cancer cell lines (Supplementary Figure 1C): have they found a general upregulation of TFEB target genes? Were TFEB-related gene ontology (GO) categories (e.g. lysosome, autophagy, etc.) enriched in the transcriptome data?

- The authors propose that TFEB controls lysosomal positioning via regulation of PI3P levels. However, most of the data to support such a claim are based on rapamycin treatment which, as stated above, does not necessarily lead to TFEB activation. Thus, the effect of rapamycin treatment on PI3P and lysosomal positioning are likely mediated by a TFEB-independent mechanism.

Furthermore, to explore the role of TFEB in PI3P regulation, the authors analyze EEA1, which is a well-known marker of early endosomes and that therefore cannot be used as an indicator of PI3P formation on ER-lysosome contact sites (where the protrudin-FYCO1 pathway has been shown to work).

A special caution has to be taken in concluding that PI3P levels change using transiently transfected probes since the brightness of the EEA1-associated signal depends on the level of the expressed probe. A normalization should be performed to the total EGFP-FYVE signal.

Finally, there is no functional characterization of the lysosomal phenotype, thus it remains unclear to what extent the lysosome dispersal is relevant for bladder cancer growth or for their invasiveness.

Reviewer #2 (Remarks to the Author):

Mathur et al. report on disrupted lysosome positioning in a variety of bladder cancer cell lines, with peripheral lysosome dispersal appearing as a hallmark of aggressive malignancy. The authors show that the master lysosomal transcriptional regulator, transcription factor EB (TFEB), has a predominantly nuclear localization in the high-grade compared to low-grade cells and that loss of TFEB reverses the lysosome dispersal phenotype in high-grade cells. Finally, the authors demonstrate that the nuclear translocation of TFEB increased endo-lysosomal PI3P levels that are important for recruitment of PI3P-binding proteins that function in anterograde lysosomal transport, such as protrudin.

Overall, this study will be of interest to the fields of organelle/lysosome trafficking, cancer cell biology and cancer metabolism. The manuscript is well written and is a timely addition to the growing literature connecting altered lysosome positioning to cancer. On this note, peripheral lysosome distribution in cancer cells has been previously reported (PMID 33110168, PMID 27105540, PMID 32018154) and these studies should be referenced in the manuscript. Finally, the link between TFEB activation and endo-lysosomal PI3P levels is lacking, thus preventing a full mechanistic understanding of the observed phenotypes.

I have a number of comments regarding the study that should be addressed:

1) Because the data suggest that a TFEB-controlled transcriptional program alters endo-lysosomal PI3P levels (and this being the signal for recruitment of protrudin which ultimately causes lysosome dispersal), the logical thing to test is whether expression of the lipid kinase, VPS34, is under the control of TFEB. Yet, the manuscript falls short of addressing this question, which could round out the entire study.

The authors do state in the discussion that, to date, it has not been demonstrated that VPS34 expression is under the control of TFEB. However, this should really be tested in their hands on the high-grade cells and compared to NHU. The authors could conduct qPCR to measure the mRNA levels of VPS34, and/or the lipid phosphatases (myotubularins) responsible for PI3P turnover. Protein levels could alternatively be measured by western blot if antibodies are available.

2) Lines 174-176, Figure S2D,E: The authors indicate that the drug U18666 was used to activate mTORC1. Looking at the reference provided by the authors that U18666a activates mTOR (Davis et al., 2021), no experiment within the paper could be found that supports this statement. In fact, other studies have shown that U18666a inhibits mTOR (PMID 20176935, see Figure 4) and the study by Willett et al shows that U18666a induces the nuclear translocation of TFE3/TFEB family of transcription factors, also suggesting that it is inhibiting mTOR (PMID 29146937, see Figure 7C).

This is important as the authors show no effect of U18666a on nucleus-to-cytosol translocation of TFEB in high-grade cells. Yet, if done properly, mTORC1 activation could reverse the high-grade phenotypes - TFEB localization and lysosome positioning. Thus, the authors should include a phospho-S6K blot to demonstrate that U18666a does indeed activate mTOR. Alternatively, a different mTORC1 activator (if one exists) could be used instead and the activity demonstrated by western blot.

3) Throughout the manuscript the experiments are conducted on high-grade cancer cells or on low-grade cells, with occasional comparisons between the two. Because the NHU condition is not tested beyond Figure 1, it is difficult to know which observations are due to normal cell biology or unique to the highly transformed line being assessed within each panel. For example, it would be interesting (and potentially useful to a wide audience) to know the following:

- a. Figure 2A-E – How much lysosomal mTOR is present in NHU cells? What is the basal mTOR activity in NHU cells? What proportion of TFEB is nuclear in NHU cells?
- b. Figure 3G – What does protrudin/lysosome staining look like in NHU cells? i.e. is the degree of protrudin recruitment to lysosomes in these cells, which have peripherally dispersed lysosomes, higher than NHU?
- c. Figure 4A-D – What is the basal EGFP-FYVE localization in NHU cells?

Minor comments/suggestions:

- 1) When looking at lysosomes across the different cell lines (Figure 1), did the authors consider doing this experiment live with a live-cell fluorescent lysosomal marker (i.e. LysoTracker or fluorescent dextran)? Some interesting morphological features of lysosomes like tubulation and motility may have been overlooked as they cannot be assessed in fixed cells (although tubulation could be preserved to some extent under specific fixation conditions). If conducted live, these experiments could be even more informative.
- 2) The authors demonstrate that the invasiveness of the individual cancer cell lines correlates with the degree of lysosome dispersal. Are those cell lines more invasive because of their lysosome phenotype? Can the authors reverse the lysosome dispersal (ie. by silencing Arl8) at least in one of the cell lines to check if this then affects invasiveness?
- 3) The authors should discuss how a number of their findings fit within the current literature:
 - a. For example, Korolchuk et al (PMID: 21394080) show that peripheral lysosomes have active mTOR. The high-grade cells tested here have low mTORC1 activity, yet their lysosomes are highly peripheral. Is this attributed to a general alteration in metabolic state in these cells? (although the published work was also conducted in a cancer cell line, HeLa).
 - b. The same study (PMID: 21394080) also showed no change in lysosome positioning with rapamycin treatment. How do the authors reconcile their results which show peripheral lysosomal dispersal upon rapamycin treatment?
 - c. Willett et al (PMID: 29146937) show that activating TFEB (nuclear translocation) upregulates TMEM55B, a dynein adaptor that causes strong perinuclear lysosomal clustering. Yet the current study reports an opposite effect on lysosome positioning as a result of TFEB nuclear translocation. Have the authors measured TMEM55B expression in the cancer cell lines? Shouldn't this dynein adaptor be upregulated in the high-grade cells that have a high proportion of nuclear TFEB? The stimuli used to activate TFEB in the study by Willett et al are different from the one used in the current study (rapamycin) so this could be the reason for different phenotypes, but this should all be discussed by the authors.
- 4) Figure 1E: How was the "average volume of lysosomes" measured and what are the units? The Y-axis says "in pix", does that refer to pixels? If so, it is not clear how volume data can be derived from pixels which are 2D.
- 5) Figure 2F,G: There is no description of what "control" is referring to. Is this untreated cells? Or were they treated with DMSO or other solvent that the drugs were reconstituted in?
- 6) Figure S2C: How significant is the increase in RAGD and TSC1 expression in high-grade cells vs. low-grade?
- 7) Figure 3G,I: It is sometimes not clear what the arrowheads are pointing to because they are small equilateral triangles. I would recommend using arrows or longer arrowheads (i.e. isosceles triangles) instead.
- 8) Although not critical because it's been previously described, but it would be nice if a positive control for the PI3P probe EGFP-FYVE, such as VPS34 inhibitor, was used to show that the probe is specific (falls off endo-lysosomes with VPS34 inhibition).
- 9) Figure S3F: What is the knockdown efficiency? The western blot should contain TFEB immunoblot as well (i.e. it should look like Figure S4I).
- 10) Line 265: The figure is called out as "Figure 3E,F", but it should say "Figure 4G-I", I believe.
- 11) Line 147: "metabolisms" should be singular "metabolism"
- 12) Line 293: "let" should say "led"
- 13) Line 545: should say "for all experiments"

Reviewer #3 (Remarks to the Author):

In the present work, Mathur and colleagues report evidences that TFEB regulates Pi3P levels on endosomal membranes which alter lysosome positioning in bladder cancer. Overall, this work is interesting, timely and well performed. However some points need to be addressed to strength the conclusions.

Major points:

1. The authors found an increase of TFEB in the nucleus, but no change in lysosome numbers. This is a discrepancy with the literature. First, the authors should utilize their transcriptomic analysis to clearly state whether TFEB downstream genes involved in lysosome biogenesis are changed between the different conditions. Second, information about the number and size of lysosomes should be provided for all experiments (as an example this is missing for figure 3C-D).
2. The recruitment of Protrudin on lysosomes is only image-based. Because the Protrudin signal is sharp, the authors should confirm these results using another complementary approach as an example by performing Western blot on purified lysosomes.
3. Image analysis of figure 4E uncovers a potential problem. The authors estimated the number of EEA1 spots over total EEA1 signal. In the vast majority of cases this should not be a problem. However, looking at the wortmannin images, a strong signal appears. This signal, in my opinion, is not corresponding to the cytosolic pool of EEA1, but rather a strong clustering of vesicles forming a large structure that in the analysis performed by the authors will be counted as 1 structure. Which artificially decreases the EEA1 signal. Maybe a higher resolution (PALM/STED/SRRF) could permit to identify individual EEA1 vesicles?
4. The authors often refer to "activation of TFEB" when using rapamycin. This is not correct since rapamycin is not a specific TFEB drug but rather acts on mTOR. This should be corrected to avoid misinterpretation of the data.
5. Could the authors exclude the possibility that decreasing EE biogenesis could impact on the effect observed on lysosomal positioning? Indeed, the centripetal lysosomal maturation could be impacted by a reduction in EE number.
6. The authors claim that little is known about lysosomal positioning in cancer cells. But a recent paper refers that an ubiquitin-like protein, UBTD1, is involved in lysosomal positioning in prostate cancer cells, and is correlated with cell proliferation and EGFR signaling. Although the referee agrees that only a limited amount of work is done on this subject, the authors should cite these studies and discuss them. Either underlining a possible link or suggesting an alternative mechanism could be useful.

Minor points:

1. Some attention should be given to removing small typographical mistakes like the lack of space before line 80 on page 4.

Reviewers' comments and point by point response:

Reviewer #1 (Remarks to the Author):

The manuscript reports a distinctive lysosome phenotype of bladder cancer cells consisting in peripheral lysosome positioning. This phenotype is accompanied by downregulated mTOR signalling and nuclear translocation of overexpressed GFP-TFEB. TFEB silencing reverts the lysosome scattering and also reduces the amount of LAMP1-associated protrudin. The authors also report that bladder cancer cells exhibit higher endosome association of the transiently overexpressed PI3P probe EGFP-FYVE domain and that this increased association is mimicked by TFEB overexpression and blunted by TFEB depletion.

While the authors identify a distinctive phenotype that could be used “as a potential biomarker for bladder cancer” they do not provide mechanistic insights linking TFEB activity to lysosome positioning and to PI3P production.

We thank the referee for their insightful and constructive suggestions. We now show that class III PI3 kinase (PIK3C3/VPS34) is under the regulation of TFEB providing the mechanistic link between TFEB activity, lysosome positioning and PI3P production. Additionally, we have investigated the role of TRPML1 in rapamycin induced TFEB translocation and clarify the implication of TFEB in recruitment of both protrudin and EEA1. Based on our results, we clarify that lysosome positioning is controlled by PI3P production that is under transcriptional control of TFEB (in addition to other mechanisms). We hope that our improved manuscript convincingly demonstrates cellular mechanisms how TFEB activity is linked to peripheral lysosome positioning in bladder cancer cells.

Specific comments:

1) The data showing TFEB activation in high-grade bladder cancer cell lines have been performed upon TFEB overexpression only. The analysis of endogenous TFEB in these cells is needed as TFEB overexpression may lead to artifacts. For instance, TFEB is a well-known rapamycin-insensitive mTORC1 substrate (PMID: 21617040, 22343943, 22692423). Thus, the data showing that rapamycin treatment results in TFEB nuclear translocation (Figure 2) are puzzling and are in contrast with previously published and well-established results.

We provide new data on endogenous TFEB localization in high-grade JMSU1 and low-grade RT112 bladder cancer cell lines that are consistent with TFEB-EGFP overexpression studies. We have added them in the new figure (**Fig. 3C, D**) and updated figure legend, methods section (page 26) and the results section page 8 line 169:

“To confirm the results of overexpressed TFEB-EGFP, we performed cell fractionation and separated the nuclear and cytosolic fractions, in which endogenous TFEB was monitored by western blotting. We used LaminB, a component of the nuclear envelope, as the nuclear marker and GAPDH as the cytosolic marker to ensure the efficiency of fractionation. Consistent with the TFEB-GFP results, we observed that TFEB was predominantly found in the nucleus in JMSU1 cells but not in RT112 cells (Fig. 3C, D). Notable, nuclear TFEB appeared to have a higher molecular weight than cytosolic TFEB in bladder cancer cell lines.”

We agree that TFEB has been shown to be a rapamycin-insensitive mTORC1 substrate in several studies. However, other studies have shown that Rapamycin can cause TFEB nuclear translocation in cells (Martina et al., 2012, Peña-Llopis et al., 2011). Indeed, we found that rapamycin induced nuclear translocation of TFEB in both low-grade cell lines, RT112 (shown in manuscript, **Fig 3F**) and MGHU3 cells (see plot below, not included in manuscript).

We provide new data for RT112 in starvation (culture in EBSS, Earl's Balanced Saline Solution), an alternative condition that leads to TFEB nuclear translocation, that are consistent with rapamycin data (TFEB-EGFP translocated to nucleus, lysosomal dispersion to periphery and more recruitment of FYVE domain proteins protrudin and EEA1 to LAMP1 positive lysosomes). Results on starvation are added to updated **Fig. S3B, C** and **Fig. S4A,F,G** and **Fig. S5C,D,G** (lower panel), the figure legends and are highlighted in the results section on page 9 line 189 and page 10 line 214, 216, 234 and page 12 line 271: "rapamycin treatment or starvation". The rapamycin dependent regulation of TFEB has been further addressed and discussed (see in more details below).

Supplementary Figure 4

Supplementary Figure 5

Martina, J.A., Chen, Y., Gucek, M., and Puertollano, R. (2012). mTORC1 functions as a transcriptional regulator of autophagy by preventing nuclear transport of TFEB. *Autophagy* 8, 903–914. <https://doi.org/10.4161/auto.19653>. PMID: 22576015

Peña-Llopis, S., Vega-Rubin-de-Celis, S., Schwartz, J.C., Wolff, N.C., Tran, T.A.T., Zou, L., Xie, X.-J., Corey, D.R., and Brugarolas, J. (2011). Regulation of TFEB and V-ATPases by mTORC1. *EMBO J.* 30, 3242–3258. <https://doi.org/10.1038/emboj.2011.257>. PMID: 21804531

2) Rapamycin can activate TRPML1 in an mTOR independent manner (PMID: 31112550). Thus, the possibility exists that in using rapamycin the authors are activating TRPML1. Indeed, from the experimental evidence presented, it looks as though mTORC1 is weakly affected, and therefore a role of calcium-TRPML1-CaN might represent a better target to study this lysosomal phenotype. TRPML1 was reported to regulate PI3P production (PMID: 31822666). This observation together with the author's data might suggest that this pathway may play a major role in TFEB regulation in the bladder cancer cells.

We agree with this comment and thank the reviewer for pointing out this insightful interpretation of our results. We have now tested the role of TRPML1 in RT112 cells. Surprisingly, inhibiting TRPML1/mucolipin-1 by ML-S11 did not prevent nuclear translocation of TFEB upon rapamycin treatment in RT112 cells. This data has been included in updated figure 3 (**Fig. 3F, G**), the figure legend and in the results section page 9 line 191:

“Rapamycin-sensitivity of TFEB and nuclear translocation was previously observed in cells with highly expressed lysosome calcium channel, mucolipin-1, also known as transient receptor potential channel mucolipin 1 (TRPML1 or MCOLN1) (Zhang et al., 2019). Rapamycin can directly activate mucolipin-1 (Zhang et al., 2019; Gan et al., 2022) that results in TFEB dephosphorylation by calcium-dependent activation of the phosphatase calcineurin (Medina et al., 2015). However, inhibition of mucolipin-1 using GW-405833 (ML-SII, 2h) in the presence of rapamycin in RT112 cells did not prevent nuclear translocation of TFEB-GFP (Fig. 3F, G).”

However, we agree that TRPML1 plays an important role in bladder cancer cell lines, because its inhibition in high-grade JMSU1 cells did prevent nuclear translocation of TFEB (Fig. 3H, I) and PI3P production in low-grade RT112 cells could potentially rely on TRPML1 (please see further clarification in our responses to comment 5 below). This has been discussed on page 14 line 326:

“we found that TFEB nuclear translocation was induced by rapamycin in RT112 cells. TFEB has been shown to be a rapamycin-insensitive mTORC1 substrate in several seminal studies (Roczniak-Ferguson et al., 2012; Settembre et al., 2011, 2012). Other studies have shown that rapamycin can cause TFEB nuclear translocation in cells (Martina et al., 2012; Peña-Llopis et al., 2011), which could be due to rapamycin induced activation of mucolipin-1 (TRPML1) (Zhang et al., 2019; Gan et al., 2022) and subsequent activation of calcineurin (Medina et al., 2015). However, inhibiting mucolipin-1 by ML-SII did not prevent nuclear translocation of TFEB upon rapamycin treatment in RT112 cells. Thus, it is not clear by which molecular mechanisms rapamycin-dependent nuclear translocation of TFEB is sustained.”...“ Because, inhibition of mucolipin-1 (TRPML1) or chelating Ca²⁺ prevented nuclear translocation of TFEB in JMSU1 cells, TFEB seems to be under the control of the mucolipin-1/calcineurin axis in these cells. Together our results indicate that the calcium/ mucolipin-1/ calcineurin axis seems to be an important target to study the lysosomal phenotype in bladder cancer.”

Zhang, X., Chen, W., Gao, Q., Yang, J., Yan, X., Zhao, H., Su, L., Yang, M., Gao, C., Yao, Y., et al. (2019). Rapamycin directly activates lysosomal mucolipin TRP channels independent of mTOR. *PLOS Biol.* 17, e3000252. <https://doi.org/10.1371/journal.pbio.3000252>. PMID: 31112550

Medina, D.L., Di Paola, S., Peluso, I., Armani, A., De Stefani, D., Venditti, R., Montefusco, S., Scotto-Rosato, A., Prezioso, C., Forrester, A., et al. (2015). Lysosomal calcium signalling regulates autophagy through calcineurin and TFEB. *Nat. Cell Biol.* 17, 288–299. <https://doi.org/10.1038/ncb3114>. PMID: 25720963

Roczniak-Ferguson, A., Petit, C.S., Froehlich, F., Qian, S., Ky, J., Angarola, B., Walther, T.C., and Ferguson, S.M. (2012). The transcription factor TFEB links mTORC1 signaling to transcriptional control of lysosome homeostasis. *Sci. Signal.* 5, ra42. <https://doi.org/10.1126/scisignal.2002790> PMID: 22692423

Settembre, C., Di Malta, C., Polito, V.A., Arencibia, M.G., Vetrini, F., Erdin, S., Erdin, S.U., Huynh, T., Medina, D., Colella, P., et al. (2011). TFEB Links Autophagy to Lysosomal Biogenesis. *Science* 332, 1429–1433. <https://doi.org/10.1126/science.1204592>. PMID: 21617040

Settembre, C., Zoncu, R., Medina, D.L., Vetrini, F., Erdin, S., Erdin, S., Huynh, T., Ferron, M., Karsenty, G., Vellard, M.C., et al. (2012). A lysosome-to-nucleus signalling mechanism senses and regulates the lysosome via mTOR and TFEB: Self-regulation of the lysosome via mTOR and TFEB. *EMBO J.* 31, 1095–1108. <https://doi.org/10.1038/emboj.2012.32>. PMID: 22343943

Martina, J.A., Chen, Y., Gucek, M., and Puertollano, R. (2012). MTORC1 functions as a transcriptional regulator of autophagy by preventing nuclear transport of TFEB. *Autophagy* 8, 903–914. <https://doi.org/10.4161/auto.19653>. PMID: 22576015

Peña-Llopis, S., Vega-Rubin-de-Celis, S., Schwartz, J.C., Wolff, N.C., Tran, T.A.T., Zou, L., Xie, X.-J., Corey, D.R., and Brugarolas, J. (2011). Regulation of TFEB and V-ATPases by mTORC1. *EMBO J.* 30, 3242–3258. <https://doi.org/10.1038/emboj.2011.257>. PMID: 21804531

Gan, N., Han, Y., Zeng, W., Wang, Y., Xue, J., & Jiang, Y. (2022). Structural mechanism of allosteric activation of TRPML1 by PI(3,5)P2 and rapamycin. *Proceedings of the National Academy of Sciences of the United States of America*, 119(7), e2120404119. <https://doi.org/10.1073/pnas.2120404119>. PMID: 35131932

3) Furthermore, the data in Supplementary Figure 2C show a very mild upregulation of two TFEB-target genes (RRAGD and TSC1), indicating little or no activation of endogenous TFEB in bladder cancer cell lines. The authors performed transcriptome analyses in several bladder cancer cell lines (Supplementary Figure 1C): have they found a general upregulation of TFEB target genes? Were TFEB-related gene ontology (GO) categories (e.g. lysosome, autophagy, etc.) enriched in the transcriptome data?

We provide new results from a Gene Set Enrichment Analysis (GSEA) to investigate the upregulation of TFEB target genes. We find that the TFEB-targeted Coordinated Lysosomal Expression and Regulation (CLEAR) network genes are higher expressed in high-grade than in low-grade bladder cancer cell lines. This enrichment supports the hypothesis that TFEB is active in high-grade cell lines. These results have been added to updated figure 3 (Fig. 3E), the figure legend, the methods section (page 27) and in the results section page 8 line 178:

“we performed a GSEA (Gene Set Enrichment Analysis) of the transcriptome of bladder cancer cell lines. GSEA analysis demonstrated that genes belonging to the TFEB-regulated Coordinated Lysosomal Expression and Regulation (CLEAR) network (Sardiello et al., 2009; Lachmann et al., 2010) are upregulated in cell lines representing high-grade bladder cancers as compared to cell lines representing low-grade bladder cancers (Fig. 3E) supporting the hypothesis that TFEB is active.”

E

Figure 3

Because TFEB is a major regulator of autophagy, we have additionally performed western blot analysis of LC3 I and II under standard growth conditions (Control, DMSO) and after Bafilomycin A treatment to monitor autophagic flux in bladder cancer cells (see below). Although total LC3 levels were comparable between different cell lines, BafA treatment strongly increased LC3-II bands indicative of

increased autophagic flux in high-grade cells consistent with a TFEB activation. We do not plan to add these results in the manuscript as it is out-of-scope from our study.

Sardiello, M., Palmieri, M., di Ronza, A., Medina, D.L., Valenza, M., Gennarino, V.A., Di Malta, C., Donaudo, F., Embrione, V., Polishchuk, R.S., et al. (2009). A Gene Network Regulating Lysosomal Biogenesis and Function. *Science* 325, 473–477. <https://doi.org/10.1126/science.1174447>. PMID: 19556463

Lachmann, A., Xu, H., Krishnan, J., Berger, S. I., Mazloom, A. R., & Ma'ayan, A. (2010). ChEA: transcription factor regulation inferred from integrating genome-wide ChIP-X experiments. *Bioinformatics (Oxford, England)*, 26(19), 2438–2444. <https://doi.org/10.1093/bioinformatics/btq466>. PMID: 20709693

4) The authors propose that TFEB controls lysosomal positioning via regulation of PI3P levels. However, most of the data to support such a claim are based on rapamycin treatment which, as stated above, does not necessarily lead to TFEB activation. Thus, the effect of rapamycin treatment on PI3P and lysosomal positioning are likely mediated by a TFEB-independent mechanism.

We do not agree that the data in our original manuscript were mostly based on rapamycin- dependent TFEB activation. Most of the PI3P data are from high-grade JMSU1 cells and siRNA-mediated depletion of TFEB (these results are now part of updated figure 5, please see **Fig. 5A-D**).

We now show new data that TFEB directly regulates class III PI3 kinase (PIK3C3/VPS34) protein levels (the enzyme responsible for the majority of PI3P production) in high-grade JMSU1 cells providing the missing mechanistic link between TFEB and PI3P (as also asked from reviewer 2). These results are added to updated figure 5 (**Fig. 5E, F**), the figure legend and in the results section page 12 line 285:

“The majority of PtdIns3P is produced by class III PI3 kinase (PIK3C3/ VPS34) which converts phosphatidylinositol (PI) to phosphatidylinositol-3-phosphate (Burke, 2018). We therefore investigated whether TFEB could regulate protein expression of PIK3C3. We found that siRNA-mediated TFEB depletion significantly decreased protein levels of PIK3C3 (Fig. 5E, F).”

Figure 5

We however agree that rapamycin treatment could lead to TFEB-independent effects through TRPML1-dependent PI3P production (PMID: 31822666) in low-grade RT112 cells as mentioned before. Therefore, we have now investigated the specific role of TFEB in increasing PI3P levels after rapamycin treatment in RT112 cells by targeting TFEB by siRNA. We find that **TFEB knockdown prevented**

the rapamycin-induced membrane binding of EEA1. Surprisingly, we however did not see the same results for protrudin. Recruitment of protrudin was TFEB-independent and could result from TRPML1-dependent PI3P production (further discussed in the next point). The results on EEA1 have been added in updated supplementary figure 5 (Fig. S5E, F), figure legends and in the results section on page 12 line 272 (for results on protrudin, see below):

“Interestingly and different to protrudin, this upregulation was dependent on TFEB, because knock down of TFEB prior to rapamycin treatment abolished the increase of EEA1 on endomembranes (Fig. S5E,F). Basal level of EEA1 recruitment on endosomes was also decreased in RT112 upon siTFEB (without any additional treatment) (Fig. S5E,F).”

Supplementary Figure 5

Total EEA1 protein levels did not show major changes siTFEB. These results are shown in updated supplementary figure 5 (Fig S5H) and in the results section on page 12 line 280:

“Total EEA1 protein levels did not majorly change under all tested conditions (Fig. S5G,H)”

Supplementary Figure 5

We provide the western blots for these analyses here which are not added to the supplementary figures due to space constraint.

Burke, J.E. (2018). Structural Basis for Regulation of Phosphoinositide Kinases and Their Involvement in Human Disease. *Mol. Cell* 71, 653–673. <https://doi.org/10.1016/j.molcel.2018.08.005>. PMID: 30193094

Scotto Rosato, A., Montefusco, S., Soldati, C., Di Paola, S., Capuozzo, A., Monfregola, J., Polishchuk, E., Amabile, A., Grimm, C., Lombardo, A., et al. (2019). TRPML1 links lysosomal calcium to autophagosome biogenesis through the activation of the CaMKK β /VPS34 pathway. *Nat. Commun.* 10, 5630. <https://doi.org/10.1038/s41467-019-13572-w>. PMID: 31822666

5) Furthermore, to explore the role of TFEB in PI3P regulation, the authors analyze EEA1, which is a well-known marker of early endosomes and that therefore cannot be used as an indicator of PI3P formation on ER-lysosome contact sites (where the protrudin-FYCO1 pathway has been shown to work).

To clarify the role of TFEB in the recruitment of protrudin we have now targeted TFEB by siRNA prior to rapamycin treatment in RT112 cells and monitored recruitment of protrudin. We find that recruitment of protrudin is TFEB-independent in RT112 cells. These results are added to updated figure 4 (Fig 4G, H), figure legends and in the results section on page 10 line 233:

“We found increased recruitment of protrudin upon both rapamycin treatment or starvation in RT112 cells (Fig. 4G,H and S4F,G). However, this recruitment was TFEB-independent, because targeting TFEB by siRNA prior to rapamycin treatment or in control conditions did not decrease protrudin levels on lysosomes (Fig. 4G,H and S4H).” ... Overall, this suggested that rapamycin and starvation induced lysosomal anterograde movement through protrudin recruitment that was TFEB-independent in RT112 cells.”

Figure 4

Treatments of rapamycin, starvation or siTFEB did not affect the total protein levels of protrudin. These results are shown in updated supplementary figure 4 (Fig S4 H,I), the figure legends and in the results section on page 10 line 237: “Total protein level of protrudin did not change in RT112 cells after rapamycin treatment, starvation or when TFEB was targeted by siRNA (Fig. S4H, I).”

Supplementary Figure 4

However, as shown in the original manuscript, depletion of TFEB in JMSU1 cells significantly reduced protrudin levels on LAMP1-positive lysosomes, showing that cell lines rely on different mechanisms/pathway for PtdIns3P production. We think that production of PtdIns3P depends on several mechanisms, e.g. via TRPML1-pathway as shown by (PMID: 31822666) and via transcriptional regulation by TFEB. We think that TFEB-dependent increase of PtdIns3P is not limited to ER-lysosome contact sites (tested by protrudin) but at the entire endolysosomal system (tested by EEA1) as stated clearly in the manuscript (see page 12 line 284: “indicating a decrease of PtdIns3P in the endosomal pathway after TFEB depletion in JMSU1 cells.”; page 19 line 442: “our results uncover a novel role of TFEB in regulating PtdIns3Ps levels on endosomes”).

Considering all the results of protrudin and EEA1 in RT112 and JMSU1 cells we have modified our conclusions stating that PtdIns3P regulates lysosome positioning and have rephrased the title and abstract of our manuscript. Moreover, we have adjusted the discussion:

Page 15 line 349: “Investigating lysosome-related mechanisms, we found that lysosome positioning is controlled by PtdIns3P in bladder cancer cell lines.”

Page 15 line 362:

“PtdIns3P formation depends mainly on the class III PI3 kinase (PIK3C3/ Vps34) (Burke, 2018). Notably, VPS34 is recruited by active mucolipin-1 (TRPML1) via a cascade including calcium/calmodulin-dependent protein kinase kinase β (CaMKK β), AMP-activated protein kinase (AMPK) and ULK1 kinase (Scotto Rosato et al., 2019). Because rapamycin directly activates mucolipin-1 (Zhang et al., 2019), PtdIns3P production via this pathway could be sufficient for the binding of protrudin (Hong et al., 2017) in RT112 cells and to induce anterograde lysosome movement. Yet, our data indicates that PtdIns3P production is also transcriptionally regulated and under the control of TFEB. We show that TFEB depletion in JMSU1 cells decreased PIK3C3 (VPS34) protein levels as well as the binding of FYVE-domain-containing proteins or the PtdIns3P probe EGFP-FYVE, indicating endosomal PtdIns3P loss. Interestingly, rapamycin-induced EEA1 recruitment on endosomes in RT112 was also TFEB- dependent, in contrast to protrudin. Support for transcriptional regulation of EEA1 recruitment in these cells comes also from the fact that endosomal EEA1 was not increased in the first 4 hours after rapamycin treatment but only after 24h, and was inhibited by cycloheximide. Thus, several parallel mechanisms seem to regulate PtdIns3P production in bladder cancer cell lines that result in TFEB-dependent and independent lysosome movements.”

We thank the reviewer to have pointed out the comparison between EEA1 and protrudin. Our data support the model proposed by Scotto Rosato et al. 2019 that TRPML1 activation leads to a TFEB-dependent response for sustained increase of PI3P levels upon longer time points (>24h) but the initial, immediate increase in PI3P, potentially at sites of phagophore formation during autophagy (Scotto Rosato et al. 2019) or ER-lysosomal contact sites (rich in protrudin) is regulated by local recruitment of VPS34 by TRPML1 after activation. However, because more data are required to support this, we have not included this statement in our manuscript.

Scotto Rosato, A., Montefusco, S., Soldati, C., Di Paola, S., Capuozzo, A., Monfregola, J., Polishchuk, E., Amabile, A., Grimm, C., Lombardo, A., et al. (2019). TRPML1 links lysosomal calcium to autophagosome biogenesis through the activation of the CaMKK β /VPS34 pathway. *Nat. Commun.* 10, 5630. <https://doi.org/10.1038/s41467-019-13572-w> PMID: 31822666

Burke, J.E. (2018). Structural Basis for Regulation of Phosphoinositide Kinases and Their Involvement in Human Disease. *Mol. Cell* 71, 653–673. <https://doi.org/10.1016/j.molcel.2018.08.005> PMID: 30193094

Hong, Z., Pedersen, N.M., Wang, L., Torgersen, M.L., Stenmark, H., and Raiborg, C. (2017). PtdIns3P controls mTORC1 signaling through lysosomal positioning. *J. Cell Biol.* 216, 4217–4233. <https://doi.org/10.1083/jcb.201611073>. PMID: 29030394

Zhang, X., Chen, W., Gao, Q., Yang, J., Yan, X., Zhao, H., Su, L., Yang, M., Gao, C., Yao, Y., et al. (2019). Rapamycin directly activates lysosomal mucolipin TRP channels independent of mTOR. *PLOS Biol.* 17, e3000252. <https://doi.org/10.1371/journal.pbio.3000252> PMID: 31112550

Gan N, Han Y, Zeng W, Wang Y, Xue J, Jiang Y. Structural mechanism of allosteric activation of TRPML1 by PI(3,5)P2 and rapamycin. *Proc Natl Acad Sci U S A.* 2022 Feb 15;119(7):e2120404119. doi: 10.1073/pnas.2120404119. PMID: 35131932

6) A special caution has to be taken in concluding that PI3P levels change using transiently transfected probes since the brightness of the EEA1-associated signal depends on the level of the expressed probe. A normalization should be performed to the total EGFP-FYVE signal.

We would like to clarify that EEA1 levels were accessed by immunostaining and we did not use EEA1 overexpression in any EEA1 experiment. In case of transient EGFP-FYVE expression, we have indeed normalized the EGFP-FYVE signal on spots to the total EGFP-FYVE expression signal as mentioned in the figures and figure legends. To clarify we have added more description on page 26 line 609: “For measurements of EEA1 or EGFP-FYVE intensities on endosomes (Fig. 5C,D; S5C-F; S5I-M) the ‘intensity on spots’ was normalized to the total fluorescence (‘per cell intensity’) of the cell analyzed“. We agree that this normalization is critical to conclude on PI3P levels.

7) Finally, there is no functional characterization of the lysosomal phenotype, thus it remains unclear to what extent the lysosome dispersal is relevant for bladder cancer growth or for their invasiveness.

We find indeed that lysosome positioning regulates invasiveness in low-grade RT112 cells, with peripheral lysosomes (after siRab7) making cells more invasive than central positioning (after siArl8b) (see below). However, we found that this was not true for highly invasive JMSU1 cells. The exact mechanisms for this are currently under investigation. We decided to focus this manuscript on the upstream mechanistic aspects of lysosome positioning changes and not on the downstream consequences for bladder cancer growth and invasion. Growth and invasion are complex phenotypes that can vary as a result of different conditions, cell lines etc. and thus require a detailed investigation for a clear understanding. Indeed, we believe that our mechanistic data presented in this manuscript will allow us to understand in detail the consequences of lysosome positioning changes in invasion and its

importance for bladder cancer, that however, is unfortunately out of the scope of this manuscript and we do not want to show this data.

Reviewer #2 (Remarks to the Author):

Mathur et al. report on disrupted lysosome positioning in a variety of bladder cancer cell lines, with peripheral lysosome dispersal appearing as a hallmark of aggressive malignancy. The authors show that the master lysosomal transcriptional regulator, transcription factor EB (TFEB), has a predominantly nuclear localization in the high-grade compared to low-grade cells and that loss of TFEB reverses the lysosome dispersal phenotype in high-grade cells. Finally, the authors demonstrate that the nuclear translocation of TFEB increased endo-lysosomal PI3P levels that are important for recruitment of PI3P-binding proteins that function in anterograde lysosomal transport, such as protrudin.

Overall, this study will be of interest to the fields of organelle/lysosome trafficking, cancer cell biology and cancer metabolism. The manuscript is well written and is a timely addition to the growing literature connecting altered lysosome positioning to cancer. On this note, peripheral lysosome distribution in cancer cells has been previously reported (PMID 33110168, PMID 27105540, PMID 32018154) and these studies should be referenced in the manuscript. Finally, the link between TFEB activation and endo-lysosomal PI3P levels is lacking, thus preventing a full mechanistic understanding of the observed phenotypes.

We thank the referee for their positive evaluation and constructive suggestions to strengthen our work. We apologize for missing references that we have now added to the manuscript on page 13, line 304. We provide new data that links TFEB activation with PI3P production through the TFEB-dependent regulation of VPS34, the enzyme responsible for the majority of PI3P production in cells.

I have a number of comments regarding the study that should be addressed:

1) Because the data suggest that a TFEB-controlled transcriptional program alters endo-lysosomal PI3P levels (and this being the signal for recruitment of protrudin which ultimately causes lysosome dispersal), the logical thing to test is whether expression of the lipid kinase, VPS34, is under the control of TFEB. Yet, the manuscript falls short of addressing this question, which could round out the entire study.

The authors do state in the discussion that, to date, it has not been demonstrated that VPS34 expression is under the control of TFEB. However, this should really be tested in their hands on the high-grade cells and compared to NHU. The authors could conduct qPCR to measure the mRNA levels of VPS34, and/or the lipid phosphatases (myotubularins) responsible for PI3P turnover. Protein levels could alternatively be measured by western blot if antibodies are available.

We thank the reviewer for this constructive suggestion. We have performed western blot analysis and indeed find that TFEB directly regulates class III PI3 kinase (PIK3C3/ VPS34) protein levels in high-grade JMSU1 cells providing the missing mechanistic link between TFEB and PI3P. These results are added to updated figure 5 (Fig. 5E, F) and updated figure legend, methods section and the results section page 12 line 285:

“The majority of PtdIns3P is produced by class III PI3 kinase (PIK3C3/ VPS34) which converts phosphatidylinositol (PI) to phosphatidylinositol-3-phosphate (Burke, 2018). We therefore further investigated whether TFEB could regulate protein expression of PIK3C3. We found that siRNA-mediated TFEB depletion significantly decreased protein levels of PIK3C3 (Fig. 5E,F).”

Figure 5

2) Lines 174-176, Figure S2D, E: The authors indicate that the drug U18666 was used to activate mTORC1. Looking at the reference provided by the authors that U18666a activates mTOR (Davis et al., 2021), no experiment within the paper could be found that supports this statement. In fact, other studies have shown that U18666a inhibits mTOR (PMID 20176935, see Figure 4) and the study by Willett et al shows that U18666a induces the nuclear translocation of TFE3/TFEB family of transcription factors, also suggesting that it is inhibiting mTOR (PMID 29146937, see Figure 7C). This is important as the authors show no effect of U18666a on nucleus-to-cytosol translocation of TFEB in high-grade cells. Yet, if done properly, mTORC1 activation could reverse the high-grade phenotypes - TFEB localization and lysosome positioning. Thus, the authors should include a phospho-S6K blot to demonstrate that U18666a does indeed activate mTOR. Alternatively, a different mTORC1 activator (if one exists) could be used instead and the activity demonstrated by western blot.

We have performed a phospho-S6K blot in JMSU1 in control (DMSO) and under U18666 treatment (see below). We do not see a clear activation of mTOR activity, however, S6K phosphorylation is clearly not reduced as shown in PMID 20176935 mentioned by the reviewer. To further clarify the activation status of mTORC1, we have added new results of another classical mTORC1 substrate, which is 4EBP1 (see also minor point 3). Together our results suggest that mTORC1 is active in all bladder cancer cells despite the fact that TFEB is nuclear in high-grade cells, and TFEB translocation is regulated by mucolipin-1 (TRPML1) in these cells. As our novel results show that bladder cancer cells with nuclear TFEB show already high mTORC1 activity (see novel Fig. 2D and see minor point 3) and the regulation of mTORC1 by U18666 is not clear and will require more investigation, we propose to not present the data on U18666 but instead present more data on the status of mTORC1 activation (4EBP1 results, see minor point 3).

Davis, O.B., Shin, H.R., Lim, C.-Y., Wu, E.Y., Kukurugya, M., Maher, C.F., Perera, R.M., Ordonez, M.P., and Zoncu, R. (2021). NPC1-mTORC1 Signaling Couples Cholesterol Sensing to Organelle Homeostasis and Is a Targetable Pathway in Niemann-Pick Type C. *Dev. Cell* 56, 260-276.e7. <https://doi.org/10.1016/j.devcel.2020.11.016>. PMID: 33308480

3) Throughout the manuscript the experiments are conducted on high-grade cancer cells or on low-grade cells, with occasional comparisons between the two. Because the NHU condition is not tested beyond Figure 1, it is difficult to know which observations are due to normal cell biology or unique to the highly transformed line being assessed within each panel. For example, it would be interesting (and potentially useful to a wide audience) to know the following:

- Figure 2A-E – How much lysosomal mTOR is present in NHU cells? What is the basal mTOR activity in NHU cells? What proportion of TFEB is nuclear in NHU cells?
- Figure 3G – What does protrudin/lysosome staining look like in NHU cells? i.e. is the degree of protrudin recruitment to lysosomes in these cells, which have peripherally dispersed lysosomes, higher than NHU?
- Figure 4A-D – What is the basal EGFP-FYVE localization in NHU cells?

We agree that these are pertinent questions. We are not able to perform these experiments, because we face limitations to obtain NHU cells. We were only able to pass the NHU cells obtained (from two

patients) up to 5 times making further studies challenging. However, transcriptome analyses have been performed from NHU cells and allow us to address some questions indirectly.

We have performed a Gene Set Enrichment Analysis (GSEA) to test whether CLEAR network genes controlled by TFEB show differences between NHU, low-grade cells (RT4, MGHU3 and RT112) and high-grade cells (TCCSup, T24, JMSU1, KU-1919) used in the study. Genes were ranked according to the spearman correlation coefficient and gene expression level was correlated with the levels of aggressiveness. We find enrichment of TFEB-regulated genes correlate with the levels of aggressiveness between NHU, low and high-grade cells. This analysis supports the hypothesis that TFEB is less active in NHU cells.

Because we cannot perform further experiments in NHU cells, and our mechanistic data focuses on the understanding of lysosomal changes between low-grade and high-grade bladder cancer cells, we did not add this data into the manuscript. Instead, we provide the GSEA of TFEB regulated genes in low-grade cells (RT4, MGHU3 and RT112) as compared to high-grade cells (TCCSup, T24, JMSU1, KU-1919), in which NHU is not included. Similarly, we find that the TFEB-targeted genes are upregulated in high-grade bladder cancer cell lines supporting the hypothesis that TFEB is active in high-grade cell lines. These results have been added to updated figure 3 (**Fig. 3E**), the figure legend, methods section (page 27) and in the results section page 8 line 178:

“we performed a GSEA (Gene Set Enrichment Analysis) of the transcriptome of bladder cancer cell lines. GSEA analysis demonstrated that genes belonging to the TFEB-regulated Coordinated Lysosomal Expression and Regulation (CLEAR) network (Sardiello et al., 2009; Lachmann et al., 2010) are upregulated in cell lines representing high-grade bladder cancers as compared to cell lines representing low-grade bladder cancers (Fig. 3E) supporting the hypothesis that TFEB is active.”

F

Figure 3

Minor comments/suggestions:

1) When looking at lysosomes across the different cell lines (Figure 1), did the authors consider doing this experiment live with a live-cell fluorescent lysosomal marker (i.e. LysoTracker or fluorescent dextran)? Some interesting morphological features of lysosomes like tubulation and motility may have been overlooked as they cannot be assessed in fixed cells (although tubulation could be preserved to some extent under specific fixation conditions). If conducted live, these experiments could be even more informative.

We have performed some preliminary experiments using live-cell imaging and have not observed major differences in lysosome dynamics. We agree that a more detailed analysis is required to look at different trafficking/sorting features of lysosomes that will be performed in the future but are out of scope of this manuscript.

2) The authors demonstrate that the invasiveness of the individual cancer cell lines correlates with the degree of lysosome dispersal. Are those cell lines more invasive because of their lysosome phenotype? Can the authors reverse the lysosome dispersal (ie. by silencing Arl8) at least in one of the cell lines to check if this then affects invasiveness?

We find indeed that lysosome positioning regulates invasiveness in low-grade RT112 cells, with peripheral lysosomes (after siRab7) making cells more invasive than central positioning (after siArl8b) (see below). However, we found that this was not true for highly invasive JMSU1 cells. The exact mechanisms for this are currently under investigation. We decided to focus this manuscript on the upstream mechanistic aspects of lysosome positioning changes and not on the downstream consequences for bladder cancer growth and invasion. Growth and invasion are complex phenotypes that can vary as a result of different conditions, cell lines etc. and thus require a detailed investigation for a clear understanding. Indeed, we believe that our mechanistic data presented in this manuscript will allow us to understand in detail the consequences of lysosome positioning changes in invasion and its importance for bladder cancer, that however, is unfortunately out of the scope of this manuscript and we do not want to show this data.

3) The authors should discuss how a number of their findings fit within the current literature:
a. For example, Korolchuk et al (PMID: 21394080) show that peripheral lysosomes have active mTOR. The high-grade cells tested here have low mTORC1 activity, yet their lysosomes are highly peripheral. Is this attributed to a general alteration in metabolic state in these cells? (although the published work was also conducted in a cancer cell line, HeLa).

We have now additionally tested mTORC1 activity by monitoring the phosphorylation of 4EBP1. Our results show that 4EBP1 substrate is strongly phosphorylated in high-grade JMSU1 cells (opposite to the phospho-S6K results provided in original manuscript). Together these data show that mTORC1 is active in cell lines with peripheral as well as central lysosomes. The results of 4EBP1 western blot analysis are added in updated figure 2 and supplementary figure 2 (**Fig 2D and Fig S2B, D, E**), the figure legends and in the results section page 7 line 152:

“we tested mTORC1 activity monitoring the phosphorylation of the direct downstream substrates p70-S6 Kinase 1 (S6K1) and eIF4E Binding Protein (4EBP1) that are phosphorylated during activation of protein synthesis. We found both substrates were phosphorylated in bladder cancer cells (Fig. 2C,D), with particularly strong phosphorylation of S6K1 in MGHU3 and RT112 cells and strong phosphorylation of 4EBP1 in JMSU1 cells. Whereas total S6K1 levels were similar in all cell lines, 4EBP1 expression was also increased in JMSU1 cells (Fig. S2A,B)”

Figure 2

Supplementary Figure 2

We discuss how our findings fit current literature by Korolchuk et al (PMID: 21394080) on page 14 line 320:

“we find the mTORC1 substrate 4EBP1 is highly expressed and phosphorylated in JMSU1 cells, whereas another substrate, p70-S6K1, is highly phosphorylated in MGHU3 and RT112 cells. Previous studies in bladder cancer have shown that overexpression of 4EBP1 correlated with increased infiltration of cancer associated fibroblasts (CAFs) and resulted in poor prognosis (Du et al., 2022). Besides these alterations in phosphorylation of different substrates, our results indicate that mTORC1 is active in cell lines with peripheral as well as central lysosomes. Korolchuk et al. have shown that activation of mTORC1 by nutrients correlates with its presence on peripheral lysosomes (Korolchuk et al., 2011). Our results reveal an additional complexity, showing that lysosome positioning could potentially correlate with differential substrate phosphorylation, in addition to mTORC1 activity. Further studies will be required to fully comprehend mTORC1 regulation by lysosome positioning.”

Du, K., Zou, J., Liu, C., Khan, M., Xie, T., Huang, X., Zhang, K., Yuan, Y., and Wang, B. (2022). A Multi-Omics Pan-Cancer Analysis of 4EBP1 in Cancer Prognosis and Cancer-Associated Fibroblasts Infiltration. *Front. Genet.* 13, 845751. <https://doi.org/10.3389/fgene.2022.845751>. PMID: 35360872

Korolchuk, V.I., Saiki, S., Lichtenberg, M., Siddiqi, F.H., Roberts, E.A., Imarisio, S., Jahreiss, L., Sarkar, S., Futter, M., Menzies, F.M., et al. (2011). Lysosomal positioning coordinates cellular nutrient responses. *Nat. Cell Biol.* 13, 453–460. <https://doi.org/10.1038/ncb2204>. PMID: 21394080

b. The same study (PMID: 21394080) also showed no change in lysosome positioning with rapamycin treatment. How do the authors reconcile their results which show peripheral lysosomal dispersal upon rapamycin treatment?

We think that this discrepancy to our results could be accounted by the complex regulation of lysosome positioning by several overlapping as well as alternative pathways found in (cancer) cells. It seems that some mechanisms/pathway are predominant in different cells and additionally may act at different time scales as part of different feedback regulations:

We have discussed this on page 16 line 386:

“This complexity in the regulation of lysosome positioning by several overlapping as well as alternative pathways found in (cancer) cells could contribute to the discrepancies between studies. It seems that mechanisms/pathway are predominant in different cells and additionally may act at several time scales as part of various feedback regulations. Thus, targeting TFEB or using potent inhibitors such as rapamycin that interfere with several players of the mTORC1-TFEB axis could converge on opposite phenotypes on lysosome positioning (Medina et al., 2011; Korolchuk et al., 2011; Willett et al., 2017).”

Korolchuk, V.I., Saiki, S., Lichtenberg, M., Siddiqi, F.H., Roberts, E.A., Imarisio, S., Jahreiss, L., Sarkar, S., Futter, M., Menzies, F.M., et al. (2011). Lysosomal positioning coordinates cellular nutrient responses. *Nat. Cell Biol.* 13, 453–460. <https://doi.org/10.1038/ncb2204>. PMID: 21394080

Medina, D.L., Fraldi, A., Bouche, V., Annunziata, F., Mansueto, G., Spanpanato, C., Puri, C., Pignata, A., Martina, J.A., Sardiello, M., et al. (2011). Transcriptional Activation of Lysosomal Exocytosis Promotes Cellular Clearance. *Dev. Cell* 21, 421–430. <https://doi.org/10.1016/j.devcel.2011.07.016>. PMID: 21889421

Willett, R., Martina, J.A., Zewe, J.P., Wills, R., Hammond, G.R.V., and Puertollano, R. (2017). TFEB regulates lysosomal positioning by modulating TMEM55B expression and JIP4 recruitment to lysosomes. *Nat. Commun.* 8, 1580. <https://doi.org/10.1038/s41467-017-01871-z>. PMID: 29146937

c. Willett et al (PMID: 29146937) show that activating TFEB (nuclear translocation) upregulates TMEM55B, a dynein adaptor that causes strong perinuclear lysosomal clustering. Yet the current study reports an opposite effect on lysosome positioning as a result of TFEB nuclear translocation. Have the

authors measured TMEM55B expression in the cancer cell lines? Shouldn't this dynein adaptor be upregulated in the high-grade cells that have a high proportion of nuclear TFEB? The stimuli used to activate TFEB in the study by Willett et al are different from the one used in the current study (rapamycin) so this could be the reason for different phenotypes, but this should all be discussed by the authors.

We have revisited the expression of TMEM55B in the transcriptome analysis and find no upregulation (see plot below). Therefore, we do not think that dynein-dependent movement is upregulated in bladder cancer cells. We not plan to add these results in the manuscript as it is out-of-scope from our study.

However, we have added the missing discussion about TMEM55B on page 16 line 379: “Other studies have reported that TFEB regulated lysosome positioning (Medina et al., 2011; Willett et al., 2017). For instance, activation of TFEB in HeLa cells has been shown to upregulate the lysosomal transmembrane protein TMEM55B resulting in central clustering of lysosomes through its interaction with dynein adapter JIP4 (Willett et al., 2017). On the other hand, TFEB overexpression has been also shown to regulate lysosome docking to plasma membrane and lysosomal exocytosis by upregulating Ca²⁺ levels through activation of mucolipin-1 (TRPML1) (Medina et al., 2011).” ... “ Our results indicate that in bladder cancer cell lines mucolipin-1-dependent mechanisms are more predominant than TMEM55B-dependent ones.”

Willett, R., Martina, J.A., Zewe, J.P., Wills, R., Hammond, G.R.V., and Puertollano, R. (2017). TFEB regulates lysosomal positioning by modulating TMEM55B expression and JIP4 recruitment to lysosomes. *Nat. Commun.* 8, 1580. <https://doi.org/10.1038/s41467-017-01871-z>. PMID: 29146937

Medina, D.L., Fraldi, A., Bouche, V., Annunziata, F., Mansueto, G., Spampanato, C., Puri, C., Pignata, A., Martina, J.A., Sardiello, M., et al. (2011). Transcriptional Activation of Lysosomal Exocytosis Promotes Cellular Clearance. *Dev. Cell* 21, 421–430. <https://doi.org/10.1016/j.devcel.2011.07.016>. PMID: 21889421

4) Figure 1E: How was the “average volume of lysosomes” measured and what are the units? The Y-axis says “in pix”, does that refer to pixels? If so, it is not clear how volume data can be derived from pixels which are 2D.

We apologize for this inaccuracy. The measurement of lysosome volume is the sum of all pixels found in all 2D planes of one segmented lysosome (Z-dimension series were acquired every 0.5 μm). This value has been obtained from the segmentation program used in batch on all cells and thus represents arbitrary units (a.u.) of lysosome volume measurement. This has been modified in the figure and clarified on page 24 line 558.

5) Figure 2F, G: There is no description of what “control” is referring to. Is this untreated cells? Or were they treated with DMSO or other solvent that the drugs were reconstituted in?

Control is DMSO. This has been added to the figure legend of now Figure 3 (See **Fig 3F, G**).

6) Figure S2C: How significant is the increase in RAGD and TSC1 expression in high-grade cells vs. low-grade?

It is not possible to test for the significance, because data are from one transcriptome analysis for JMSU1 and KU19-19 cells. However, we have now performed a Gene Set Enrichment Analysis (GSEA) to test whether CLEAR network genes controlled by TFEB show differences between high-grade and low-grade cells used in the study. GSEA analysis demonstrated that CLEAR network genes are more upregulated in high-grade than in low-grade bladder cancer cells with high significance. This enrichment supports the hypothesis that TFEB is active in high-grade cells. These results have been added to figures and the manuscript as described above (see response to reviewer comment **3**).

7) Figure 3G,I: It is sometimes not clear what the arrowheads are pointing to because they are small equilateral triangles. I would recommend using arrows or longer arrowheads (i.e. isosceles triangles) instead.

This has been modified in all figures which had white arrows.

8) Although not critical because it's been previously described, but it would be nice if a positive control for the PI3P probe EGFP-FYVE, such as VPS34 inhibitor, was used to show that the probe is specific (falls off endo-lysosomes with VPS34 inhibition).

We have verified this and results have now been added to updated figure 5 (**Fig. S5A, B**), the figure legend and in the results section page 11 line 260:

“Treatment of EGFP-FYVE expressing cells with wortmannin to inhibit PtdIns3P production by phosphatidylinositol-3-phosphate kinases significantly reduced the levels of EGFP-FYVE on LAMP1-positive lysosomes confirming the specific binding of this construct to PtdIns3P on lysosomes (**Fig. S5A,B**).”

Supplementary Figure 5

9) Figure S3F: What is the knockdown efficiency? The western blot should contain TFEB immunoblot as well (i.e. it should look like Figure S4I).

The blot of TFEB has been added. This figure is now part of supplementary figure 4 (**Fig S4J**) that is shown below and updated figure legend is added in manuscript.

J

Supplementary Figure 4

10) Line 265: The figure is called out as “Figure 3E,F”, but it should say “Figure 4G-I”, I believe.

11) Line 147: “metabolisms” should be singular “metabolism”

12) Line 293: “let” should say “led”

13) Line 545: should say “for all experiments”

All the typing error have been modified.

Reviewer #3 (Remarks to the Author):

In the present work, Mathur and colleagues report evidences that TFEB regulates Pi3P levels on endosomal membranes which alter lysosome positioning in bladder cancer. Overall, this work is interesting, timely and well performed. However, some points need to be addressed to strength the conclusions.

We thank the referee for their positive evaluation and constructive suggestions to strengthen our work.

Major points:

1) The authors found an increase of TFEB in the nucleus, but no change in lysosome numbers. This is a discrepancy with the literature. First, the authors should utilize their transcriptomic analysis to clearly states whether TFEB downstream genes involved in lysosome biogenesis are changed between the different conditions. Second, informations about the number and size of lysosomes should be provided for all experiments (as example this is missing for the figure 3C-D).

We have now performed a Gene Set Enrichment Analysis (GSEA) to test whether CLEAR network genes controlled by TFEB show differences between high-grade cells (TCCSup, T24, JMSU1, KU-1919), and low-grade cells (RT4, MGHU3 and RT112) used in the study. GSEA analysis demonstrated that CLEAR network genes are more upregulated in high-grade than in low-grade bladder cancer cells. This enrichment supports the hypothesis that TFEB is active in high-grade cells. These results have been added to updated figure 3 (**Fig. 3E**), the figure legend, methods section (page 27) and in the results section page 8 line 178:

“we performed a GSEA (Gene Set Enrichment Analysis) of the transcriptome of bladder cancer cell lines. GSEA analysis demonstrated that genes belonging to the TFEB-regulated Coordinated Lysosomal Expression and Regulation (CLEAR) network (Sardiello et al., 2009; Lachmann et al., 2010) are upregulated in cell lines representing high-grade bladder cancers as compared to cell lines representing low-grade bladder cancers (Fig. 3E) supporting the hypothesis that TFEB is active.”

E

Figure 3

We have analyzed lysosome number per cell in RT112 cells after rapamycin treatment (which results in nuclear translocation of TFEB), and indeed find an increase in lysosome numbers. These results are added into supplementary figure 4 (**Fig S4B**), figure legend and in the results section page 10 line 220: “Interestingly, we also observed an increase in the number of lysosomes per cell after rapamycin treatment potentially indicating activation of lysosomal biogenesis (Fig. S4B).”

B

Supplementary Figure 4

However, our initial analysis in different bladder cancer cells did not reveal a systematic increase in lysosome numbers or size for bladder cancer cell lines (see Fig. 1D,E) for which we however find significant enrichment of CLEAR network transcripts in the Gene Set Enrichment Analysis (GSEA). Thus, we do not see a correlation between lysosome numbers/size increase and TFEB activation. TFEB activation can also lead to several alternative phenotypes, including autophagy, lysosomal exocytosis and some of these phenotypes seem to be specific to tissues and disease conditions (Napolitano and Ballabio, 2016). Our results suggest that anterograde lysosome movement could be an additional alternative phenotype that seems to be predominant in bladder cancer cells. We discuss this on page 13 line 304: “Lysosomal dispersion or scattering has been reported in cancers such as breast cancer (Wu et al., 2020), prostate cancer (Dykes et al., 2016) or hepatocellular carcinomas (Lyu et al., 2020), where this phenotype has been shown to be associated with increased cancer invasiveness. The phenotype of lysosomal dispersion is different to the previously described expansion of the lysosomal compartment in pancreatic ductal adenocarcinoma (PDA) and indicative of increased lysosome biogenesis (Perera et al., 2015). Indeed, we did not observe a systematic increase in lysosome numbers or size in cell lines representing high-grade cancers.”

Napolitano, G., and Ballabio, A. (2016). TFEB at a glance. *J. Cell Sci.* jcs.146365. <https://doi.org/10.1242/jcs.146365>. PMID: 27252382

2) The recruitment of Protrudin on lysosomes is only image-based. Because the Protrudin signal is sharp, the authors should confirm these results using another complementary approach as example by performing Westernblot on purified lysosomes.

Protrudin is an ER transmembrane protein and localizes on lysosomes only through the transient interaction with PI3P through its FYVE domain. Therefore, we do not think that protrudin can be enriched in the lysosomal fraction after purification, and the suggested experiment will not give us the desired results. The recruitment of protrudin on lysosomes is well established (Hong et al., 2017). Moreover, we argue that PI3P increase is the important downstream effect of TFEB activation and alternative proteins that bind PI3P could participate in the anterograde lysosome movement as discussed in our original manuscript (please see now on page 15 line 357)

Hong, Z., Pedersen, N.M., Wang, L., Torgersen, M.L., Stenmark, H., and Raiborg, C. (2017). PtdIns3P controls mTORC1 signaling through lysosomal positioning. *J. Cell Biol.* 216, 4217–4233. <https://doi.org/10.1083/jcb.201611073>. PMID: 29030394

3) Image analysis of the figure 4E uncovers a potential problem. The authors estimated the number of EEA1 spots over total EEA1 signal. In the vast majority of the cases this should not be a problem. However, looking at the wortmannin images, a strong signal appears. This signal, to my opinion, is not corresponding to cytosolic pool of EEA1, but rather a strong clustering of vesicles forming a large structure that in the analysis performed by the authors will be count as 1 structure. Which artificially

decrease the EEA1 signal. May be a higher resolution (PALM/STED/SRRF) could permit to identify individual EEA1 vesicles?

We did not estimate the **numbers** of EEA1 spots but measured the **intensity** of EEA1 spots over the intensity of total EEA1 signal per cell. Therefore the clustering will not provide a problem for the analysis. Anyway, we have verified the analysis and the signal is not identified as one structure but as several ones. To clarify we have modified the axis label and added more description on the measurement on page 26 line 609: “For measurements of EEA1 or EGFP-FYVE intensities on endosomes (Fig. 5C,D; S5C-F; S5I-M) the ‘intensity on spots’ was normalized to the total fluorescence (‘per cell intensity’) of the cell analyzed“. This figure is now part of supplementary figure 5 (please see **Fig. S5 L, M**)

4) The authors often refer as “activation of TFEB” when using rapamycin. This is not correct since rapamycin is not a specific TFEB drug but rather acts on mTOR. This should be corrected to avoid misinterpretation of the data.

We agree and thank the referee for pointing out this. This has been modified throughout the text.

5) Could the authors exclude the possibility that decreasing EE biogenesis could impact on the effect observed on lysosomal positioning? Indeed, the centripetal lysosomal maturation could be impacted by a reduction in EE number.

We do not exclude this. Indeed, PI3P is important for endosomal maturation, and the upregulation of PI3P by TFEB could potentially regulate endosomal maturation in addition to lysosomal biogenesis. However, we have tested whether EGFR degradation is different in RT112 and JMSU1 cells showing differential lysosome position and found this is **not** the case (see below). This indicates that endolysosomal maturation is comparable in different bladder cancer cell lines. We do not plan to add these results in the manuscript.

6) The authors claim that little is known about lysosomal positioning in cancer cells. But a recent paper refer that an ubiquitin-like protein, UBTD1, is involved in lysosomal positioning in prostate cancer cells, and is correlated with cell proliferation and EGFR signaling. Although the referee agrees that only a limited amount of work is done on this subject, the authors should cite these studies and discuss them. Either underlining a possible link or suggesting alternative mechanism could be useful.

We apologize for missing reference that we have now added to the manuscript on page 17, line 397: “Intriguingly, in addition to PtdIns3P, the ubiquitin system has been shown to regulate lysosome positioning, particularly the ER-embedded UBE2J1/RNF26 ubiquitylation complex and the opposing deubiquitinating enzyme USP15 (Jongsma et al., 2016; Cremer T et al., 2021), the ubiquitin domain-containing protein 1 (UBTD1) (Torrino et al., 2021) and deubiquitinase USP17 (Lin J et al. 2022). Further investigations will reveal whether these represent an alternative pathway or whether ubiquitylation cross talks with phosphoinositide regulation.”

Jongsma ML, Berlin I, Wijdeven RH, Janssen L, Janssen GM, Garstka MA, Janssen H, Mensink M, van Veelen PA, Spaapen RM, Neefjes J. An ER-Associated Pathway Defines Endosomal Architecture for Controlled Cargo Transport. *Cell*. 2016 Jun 30;166(1):152-66. doi: 10.1016/j.cell.2016.05.078.

PMID: 27368102

Cremer T, Jongsma MLM, Trulsson F, Vertegaal ACO, Neefjes J, Berlin I. The ER-embedded UBE2J1/RNF26 ubiquitylation complex exerts spatiotemporal control over the endolysosomal pathway. *Cell Rep.* 2021 Jan 19;34(3):108659. doi: 10.1016/j.celrep.2020.108659. PMID: 33472082

Torrino, S., Tiroille, V., Dolfi, B., Dufies, M., Hinault, C., Bonesso, L., Dagnino, S., Uhler, J., Irondelle, M., Gay, A., et al. (2021). UBTD1 regulates ceramide balance and endolysosomal positioning to coordinate EGFR signaling. *ELife* 10, e68348. <https://doi.org/10.7554/eLife.68348> PMID: 33884955

Lin J, McCann AP, Sereesongsaeng N, Burden JM, Alsa'd AA, Burden RE, Micu I, Williams R, Van Schaeybroeck S, Evergren E, Mullan P, Simpson JC, Scott CJ, Burrows JF. USP17 is required for peripheral trafficking of lysosomes. *EMBO Rep.* 2022 Apr 5;23(4):e51932. doi: 10.15252/embr.202051932. Epub 2022 Jan 26. PMID: 35080333

Minor points:

1) Some attention should be done to remove small typography mistakes like the lake of space line80 page4.

Typing errors have been corrected

Reviewers' comments:

Reviewer #1 (Remarks to the Author):

In the revised manuscript the authors have introduced new data in response to the reviewer comments, however some major points have not been clarified. A possible reason might be that the cellular systems (high-grade bladder cancer cell lines) used by the authors are quite unique in terms of TFEB dynamics; for this reason they would deserve a comprehensive study of the mechanisms regulating TFEB to be compared and reconciled with the current literature. Moreover, a comparative analysis performed in other cell types would also help in interpreting the results obtained with the specific cell types studied in the manuscript. Three main concerns remain unsatisfactorily addressed.

A. Endogenous TFEB activation. The authors decided to assess the nuclear translocation of endogenous TFEB by subcellular fractionation and immunoblot, and not by immunofluorescence (why?). However the data presented are rather confusing and do not solve the problem; an analysis of endogenous TFEB distribution by IF might help in solving it. Below my specific comments

1. There is no comparison with the localization of TFEB in non-cancer cells, and therefore without IF and without this control, it is difficult to conclude that endogenous TFEB is mostly localized into the nucleus in bladder cancer cells.

2. The fractionation data show that TFEB in nuclear fractions has a higher molecular weight than in cytosolic fractions (Figure 3C), as also stated by the authors. These findings are in net contrast with robust published literature showing that nuclear TFEB is de-phosphorylated, thus showing a lower molecular weight compared to cytosolic TFEB. The authors leave the reader with no explanation of their puzzling result, and this casts doubt on the conclusion and interpretation of the data. Have they considered the possibility that the upper band they follow is in fact not TFEB, while the lower one could be TFEB? Can they show a wider area of the blot (including the MW markers)? Other explanations?

B. Regulation of VPS34 by TFEB.

1. The authors suggest a transcriptional regulation of VPS34, so is VPS34 a target of TFEB? Gain and loss of function experiment modulating TFEB levels are required to conclude that TFEB controls VPS34. These experiments should look at VPS34 mRNA and not only at protein levels.

2. In fact protein levels of VPS34 show only a minor decrease (20%) upon TFEB depletion by siRNA, casting doubt that TFEB may play a major role in the control of VPS34 levels.

C. Functional relevance of the observed phenotype

It remains unclear to what extent the lysosome dispersal is relevant for bladder cancer growth or for their invasiveness. The authors felt that this is out of the scope of their manuscript. I believe this is an important point, otherwise the rationale and relevance for studying lysosomal positioning in these cancer cells remains obscure.

Minor comment

The data indicating that rapamycin treatment induces TFEB nuclear translocation are in contrast with several other reports. Please note that one of the manuscripts indicated in the authors' rebuttal as an example of rapamycin-induced TFEB nuclear translocation (Pena-Llopis et al., 2011), in fact shows the opposite, as rapamycin was shown in this paper to paradoxically promote TFEB cytosolic re-localization in TSC2-KO cells (in which TFEB shows a constitutively nuclear localization).

Reviewer #2 (Remarks to the Author):

The authors have addressed the majority of my concerns. The new data on lysosome positioning as it affects cancer cell invasiveness is particularly nice. It is quite interesting also that the expression of VPS34 was found to be decreased in the high-grade cells that had depleted TFEB

levels (siTFEB). This helps to support the claim that TFEB can control PI(3)P levels, at least in high-grade bladder cancer cell lines.

It is unfortunate that limitations in obtaining NHU cells precludes further comparison of phenotypes between the cancer cells and "normal" cells. However, I appreciate that the authors are largely comparing "high-grade" vs "low-grade" cancer cells in this work.

Overall, I am satisfied with the additional data that has been added. This study will be of great interest to a wide audience and I commend the authors for their efforts in strengthening the findings with the additional experiments.

Reviewer #3 (Remarks to the Author):

The authors replies to all my queries.

Reviewers' comments:

Reviewer #1 (Remarks to the Author):

In the revised manuscript the authors have introduced new data in response to the reviewer comments, however some major points have not been clarified. A possible reason might be that the cellular systems (high-grade bladder cancer cell lines) used by the authors are quite unique in terms of TFEB dynamics; for this reason they would deserve a comprehensive study of the mechanisms regulating TFEB to be compared and reconciled with the current literature. Moreover, a comparative analysis performed in other cell types would also help in interpreting the results obtained with the specific cell types studied in the manuscript. Three main concerns remain unsatisfactorily addressed.

It has been shown in the literature that TFEB dynamics are strongly context dependent (as highlighted by a recent review, PMID: 36231114). Although TFEB dynamics in bladder cancer cells show some discrepancy with some publications, they can be reconciled with other papers (e.g. cell lines that show high mucolipin activity) as already discussed on page 14 line 330. However, we agree that aggressive bladder cancer cell lines could show unique features due to their transformation. Indeed, in our study we were interested to identify these deregulations. While we have identified the molecular mechanisms involved in deregulation, future studies will address their relevance in patients and disease progression. To highlight that we agree on this point with the referee, **we have now added the following to the discussion on page 19 line 449**: “In conclusion, we reveal unexpected phenotypes in terms of TFEB dynamics in aggressive bladder cancer cells lines. Future studies will be required to comprehensively depict the mechanisms regulating TFEB as well as their relevance in patients and disease progression.”

Franco-Juárez B, Coronel-Cruz C, Hernández-Ochoa B, Gómez-Manzo S, Cárdenas-Rodríguez N, Arreguin-Espinosa R, Bandala C, Canseco-Ávila LM, Ortega-Cuellar D. TFEB; Beyond Its Role as an Autophagy and Lysosomes Regulator. *Cells*. 2022 Oct 7;11(19):3153. doi: 10.3390/cells11193153. PMID: 36231114

A. Endogenous TFEB activation. The authors decided to assess the nuclear translocation of endogenous TFEB by subcellular fractionation and immunoblot, and not by immunofluorescence (why?). However the data presented are rather confusing and do not solve the problem; an analysis of endogenous TFEB distribution by IF might help in solving it. Below my specific comments

First, **we apologize for the missing information about the anti-TFEB antibody and added this important information on page 23 line 559. The antibody used (CST: 4240) did not give a**

specific signal in bladder cancer cells in immunofluorescence assays (see point 1). Therefore, unfortunately, immunofluorescence-based assays did not allow us to address endogenous TFEB dynamics in bladder cancer cells.

1. There is no comparison with the localization of TFEB in non-cancer cells, and therefore without IF and without this control, it is difficult to conclude that endogenous TFEB is mostly localized into the nucleus in bladder cancer cells.

We have performed immunofluorescence against endogenous TFEB in bladder cancer cells (see images for IF analysis in RT112 cells, along with the corresponding quantification below).

Unfortunately, although the antibody (CST: 4240) showed a reduction in signal in WB analysis after TFEB knock down (see different examples below including from Fig. S4H), it only gave non-specific signals in IF, because the signal was not significantly reduced after siTFEB. This made us conclude that the Ab did not work in IF. On account of these limitations, we have used the TFEB-EGFP and immunoblot analysis to study TFEB localization in bladder cancer cells. Both of these analyses used have shown us consistent results.

We agree that it would be desirable to compare cancer cells with NHU cells, but unfortunately, we are not able to perform these experiments, because we face limitations to obtain NHU cells. We were only able to pass the NHU cells obtained (from two patients) up to 5 times making further studies challenging. However, **transcriptome analyses have been performed from NHU cells and allow us to address some questions indirectly.**

We have performed a Gene Set Enrichment Analysis (GSEA) to test whether CLEAR network genes controlled by TFEB show differences between NHU, low-grade cells (RT4, MGHU3 and RT112) and high-grade cells (TCCSup, T24, JMSU1, KU-1919) used in the study. Genes were ranked according to the spearman correlation coefficient and gene expression level was correlated with the levels of aggressiveness. We find **enrichment of TFEB-regulated genes correlate with the levels of aggressiveness between NHU, low and high-grade cells**. This analysis supports the hypothesis that TFEB is more active in aggressive cell lines (TCCSup, T24, JMSU1, KU-1919) than in less aggressive ones (RT4, MGHU3 and RT112) or NHU.

2. The fractionation data show that TFEB in nuclear fractions has a higher molecular weight than in cytosolic fractions (Figure 3C), as also stated by the authors. These findings are in net contrast with robust published literature showing that nuclear TFEB is de-phosphorylated, thus showing a lower molecular weight compared to cytosolic TFEB. The authors leave the reader with no explanation of their puzzling result, and this casts doubt on the conclusion and interpretation of the data. Have they considered the possibility that the upper band they follow is in fact not TFEB, while the lower one could be TFEB? Can they show a wider area of the blot (including the MW markers)? Other explanations?

Please see below the western blot shown in Fig. 5E (along with the corresponding membranes showing the molecular marker) of TFEB in control (siLUC) and siTFEB treatments in JMSU1 cells

(and above for RT112 cells). The blot shows that the antibody (CST: 4240) **recognizes at least two bands found at approximately 70KDa and that ALL bands are reduced upon siTFEB**. This suggests that all bands at approximately 70KDa correspond to TFEB.

We also provide below the full uncropped western blots presented in the manuscript Fig.3C (showing Lamin B, a marker of nuclear fraction; GAPDH, a marker of cytosolic fraction and endogenous TFEB) along with the corresponding membranes showing the molecular marker. The black box marks the area of the western blot show in Fig. 3C. Both bands of TFEB corresponding to a molecular weight of approximately 70KDa have been used for the analysis.

We acknowledge that as per our current understanding and publications, the nuclear TFEB is dephosphorylated and thus should have a lower molecular weight. This is the reason we actually highlighted our observation. However, nuclear export of TFEB has been shown to require mTOR dependent phosphorylation at sites S142 and S138 (whether direct or indirect remains unknown) that is proposed to occur in the nucleus (Napolitano et al., 2018; Ballabio and Bonifacino, 2020). This suggests that phosphorylated fractions of TFEB could also be found in nucleus potentially attributing to a higher molecular weight band in western blot. Importantly, in addition to phosphorylation, TFEB has been shown to be regulated by various post-translational level, including phosphorylation, acetylation, SUMOylation, PARylation, and glycosylation (reviewed in PMID: 36231114) that could in addition impact protein migration. Moreover different isoforms of TFEB have been described (Vu et al. 2021) that could be present

in different cells and differentially distributed between nucleus and cytoplasm. As we **can only speculate, we have added to the manuscript on page 8 line 177**: “This observations opens the question whether post-translational modifications, such as acetylation, SUMOylating, PARsylation, or glycosylation in addition to phosphorylation (Franco-Juárez et al., 2022), or alternatively, different isoforms of TFEB could be more prevalent in the nuclear fraction of bladder cancer cells.”

Napolitano, G., Esposito, A., Choi, H. *et al.* mTOR-dependent phosphorylation controls TFEB nuclear export. *Nat Commun* 9, 3312 (2018). <https://doi.org/10.1038/s41467-018-05862-6>

Ballabio A, Bonifacino JS. Lysosomes as dynamic regulators of cell and organismal homeostasis. *Nat Rev Mol Cell Biol.* 2020 Feb;21(2):101-118. doi: 10.1038/s41580-019-0185-4

Franco-Juárez B, Coronel-Cruz C, Hernández-Ochoa B, Gómez-Manzo S, Cárdenas-Rodríguez N, Arreguin-Espinosa R, Bandala C, Canseco-Ávila LM, Ortega-Cuellar D. TFEB; Beyond Its Role as an Autophagy and Lysosomes Regulator. *Cells.* 2022 Oct 7;11(19):3153. doi: 10.3390/cells11193153. PMID: 36231114

Hong Nhung Vu, Ramile Dilshat, Valerie Fock, Eiríkur Steingrímsson. User guide to MiT-TFE isoforms and post-translational modifications. doi.org/10.1111/pcmr.12922. *Pigment cell & melanoma research* 2021, 34(1)13-27.

B. Regulation of VPS34 by TFEB.

1. The authors suggest a transcriptional regulation of VPS34, so is VPS34 a target of TFEB? Gain and loss of function experiment modulating TFEB levels are required to conclude that TFEB controls VPS34. These experiments should look at VPS34 mRNA and not only at protein levels.

We performed additional experiments to test mRNA levels of VPS34 after siRNA-mediated depletion of TFEB and demonstrate that **mRNA levels are decreased at similar levels as protein levels**, see below and in the Fig. S5C and legends, as well as text page 13 line 293 and M&M section page 21 line 500. This further supports our claim of TFEB-dependent regulation of VPS34 in JMSU1 cells. VPS34 has been proposed to be regulated by TFEB recently (Tao et al. 2021), however it is not found in the CLEAR genes list (Sardiello et al., 2009; Lachmann et al., 2010). Therefore, further studies are needed to shed light on the molecular mechanisms of regulation. **This has been added to the text on page 17, line 401.**

Tao H, Yancey PG, Blakemore JL, Zhang Y, Ding L, Jerome WG, Brown JD, Vickers KC, Linton MF. Macrophage SR-BI modulates autophagy via VPS34 complex and PPAR α transcription of Tfeb in atherosclerosis. *J Clin Invest.* 2021 Apr 1;131(7):e94229. PMID: 33661763

2. In fact protein levels of VPS34 show only a minor decrease (20%) upon TFEB depletion by siRNA, casting doubt that TFEB may play a major role in the control of VPS34 levels.

We find that mRNA levels are decreased at similar levels as protein levels supporting our initial results. As mentioned, TFEB dynamics are strongly context dependent (PMID: 36231114) due to several feedback regulation pathways and thus compensatory mechanisms could influence the regulation. Moreover, TFEB is one of four family members with partly overlapping functions, thus compensatory mechanisms of these family members could be envisioned. To **highlight that further studies are required to comprehensively understand TFEB-dependent regulation of VPS34 we have added in the discussion on page 17 line 401:** “VPS34 has been proposed to be regulated by TFEB recently (Tao et al. 2021), however it is not found in the CLEAR genes list (Sardiello et al., 2009; Lachmann et al., 2010). Therefore, further studies are needed to shed light on the molecular mechanisms of regulation.”

C. Functional relevance of the observed phenotype

It remains unclear to what extent the lysosome dispersal is relevant for bladder cancer growth or for their invasiveness. The authors felt that this is out of the scope of their manuscript. I believe this is an important point, otherwise the rationale and relevance for studying lysosomal positioning in these cancer cells remains obscure.

The aim was to link changes in phenotype with molecular mechanisms. While we have identified the molecular mechanisms involved in deregulation, future studies will address their relevance in patients and disease progression.

Minor comment

The data indicating that rapamycin treatment induces TFEB nuclear translocation are in contrast with several other reports. Please note that one of the manuscripts indicated in the authors' rebuttal as an example of rapamycin-induced TFEB nuclear translocation (Pena-Llopis et al., 2011), in fact shows the opposite, as rapamycin was shown in this paper to paradoxically promote TFEB cytosolic re-localization in TSC2-KO cells (in which TFEB shows a constitutively nuclear localization).

The reference should have been Zhang et al., 2019. **We apologize for this mistake. We have removed the citation (Pena-Llopis et al., 2011) on page 14 line 334, as it does not support our claim, and exchange it by Zhang et al., 2019** as examples of studies showing rapamycin-induced TFEB nuclear translocation.

Zhang, X., Chen, W., Gao, Q., Yang, J., Yan, X., Zhao, H., Su, L., Yang, M., Gao, C., Yao, Y., et al. (2019). Rapamycin directly activates lysosomal mucolipin TRP channels independent of mTOR. PLOS Biol. 17, e3000252. <https://doi.org/10.1371/journal.pbio.3000252> PMID: 31112550